# Integrating images from multiple microscopy screens reveals diverse patterns of change in the subcellular localization of proteins

Alex X Lu[1], Yolanda T Chong[2], Ian Shen Hsu[3], Bob Strome[3], Louis-Francois Handfield[1], Oren Kraus[4], Brenda J Andrews[2,5], Alan M Moses[1,3,6]*

[1]Department of Computer Science, University of Toronto, Toronto, Canada; [2]Terrence Donnelly Centre for Cellular and Biomolecular Research, University of Toronto, Toronto, Canada; [3]Department of Cell and Systems Biology, University of Toronto, Toronto, Canada; [4]Department of Electrical and Computer Engineering, University of Toronto, Toronto, Canada; [5]Department of Molecular Genetics, University of Toronto, Toronto, Canada; [6]Center for Analysis of Genome Evolution and Function, University of Toronto, Toronto, Canada

**Abstract** The evaluation of protein localization changes on a systematic level is a powerful tool for understanding how cells respond to environmental, chemical, or genetic perturbations. To date, work in understanding these proteomic responses through high-throughput imaging has catalogued localization changes independently for each perturbation. To distinguish changes that are targeted responses to the specific perturbation or more generalized programs, we developed a scalable approach to visualize the localization behavior of proteins across multiple experiments as a quantitative pattern. By applying this approach to 24 experimental screens consisting of nearly 400,000 images, we differentiated specific responses from more generalized ones, discovered nuance in the localization behavior of stress-responsive proteins, and formed hypotheses by clustering proteins that have similar patterns. Previous approaches aim to capture all localization changes for a single screen as accurately as possible, whereas our work aims to integrate large amounts of imaging data to find unexpected new cell biology.
DOI: https://doi.org/10.7554/eLife.31872.001

*For correspondence:
alan.moses@utoronto.ca

**Competing interests:** The authors declare that no competing interests exist.

## Introduction

The ability to control the subcellular localization of proteins has long been understood as an important component of a cell's regulatory toolkit in response to perturbations (*Cyert, 2001*; *Bauer et al., 2015*; *Protter and Parker, 2016*), such as drug treatments, genetic mutations, or environmental stressors. Towards the goal of systematically characterizing these proteome dynamics, high-throughput technologies have been employed to gather data about protein localization in cells (*Yuet and Tirrell, 2014*; *Dephoure and Gygi, 2012*; *Nagaraj et al., 2012*). Here, we focus on the analysis of images from screens of libraries of yeast strains expressing green fluorescent protein (GFP)-tagged proteins generated using automated high-throughput microscopy (*Mattiazzi Usaj et al., 2016*; *Caicedo et al., 2016*). These experiments yield terabyte-scale image datasets (*Koh et al., 2015*; *Riffle and Davis, 2010*; *Breker et al., 2014*) that show changes in the subcellular localization of the proteome in response to varied perturbations (*Chong et al., 2015*; *Tkach et al., 2012*; *Kraus et al., 2017*; *Breker et al., 2013*).

**eLife digest** Cells are busy structures in which proteins move from one place to the other to perform their roles. To survive, organisms must constantly adapt to changes, such as mutations, new environments, or encountering drugs. Often, these adaptive responses involve groups of proteins traveling to new locations in the cell. Tracking these movements is useful to understand which biological processes cells activate to cope with modifications.

To study these mechanisms, scientists conduct experiments where they expose cells to one type of change – for example they mutate one gene, or give one drug. They then follow the proteins in the cell using powerful microscopes.

These instruments can take thousands of pictures of cells every day, which results in large amounts of data. The images are then analyzed, which is often subjective, time-consuming or cannot easily be repeated on other experiments. This prevents researchers from considering more than one experiment at the time, and it makes it difficult to compare how cells respond across many different types of changes.

Here, Lu et al. combine and analyze 400,000 images from 24 experiments conducted on yeasts; they use a computational method that automatically measures the differences in the locations of the proteins between experiments. This new large-scale approach reveals aspects of cell biology that are not obvious from the results of any one experiment alone. For example, responses that are specific to one type of change can be distinguished from the ones that occur each time cells are exposed to new conditions. It also becomes possible, for the first time, to group together proteins that move in the same way across different types of changes, and to infer their roles. For instance, a protein was shown to be 'pulsing' – moving quickly between two specific compartments in the cell – based on how it shared certain features with other pulsing proteins.

This computational approach is not limited to experiments on yeasts, and could be used on studies in human cells as well. Drug companies could then compare at a large scale how different treatments affect protein localization.

DOI: https://doi.org/10.7554/eLife.31872.002

The growing quantity and diversity of imaging experiments examining protein localization in yeast open the possibility of integrating proteomic information from different image screens. Integrative strategies are widely appreciated for the analysis of high-throughput transcript profiles, such as RNA-seq or microarray data, providing mutual context and hypothesis discovery that would have not been evident in independent analyses of the datasets (reviewed in *Dolinski and Troyanskaya, 2015*). One well-known example is the analysis of the environmental stress response in yeast; by combining the results of 142 different microarray experiments, *Gasch et al. (2000)* identified a large set of genes whose transcript levels changed under most environmental stress perturbations as part of a general program. However, transcriptome profiling experiments capture only part of the overall biomolecular activity in a cell. To extend these highly successful strategies for genomic data to a wider perspective, researchers are also interested in integrating information about the proteome. Protein interaction (*Greene et al., 2015*) and protein expression data (*Cenik et al., 2015*) have been integrated to reveal insights about tissue- and individual-specific regulatory variation in humans, but the integration of data about subcellular protein localization changes is rare despite the important regulatory role of these changes. Although protein localizations can be observed from fluorescent cell micrographs and have been catalogued qualitatively in databases such as the Human Cell Atlas (*Thul et al., 2017*) and Uniprot (*UniProt Consortium, 2015*), the automated comparison of localization changes from these images is challenging.

To integrate localization patterns from multiple screens, scalable data acquisition and analysis methods that produce consistent and quantitative summaries of protein localization changes are required. Using automated fluorescence microscopy, images can be rapidly acquired from many different yeast cell cultures grown in defined growth conditions (*Chong et al., 2015*; *Tkach et al., 2012*; *Kraus et al., 2017*). However, to date, analysis methods based on visual inspection or supervised machine learning have been applied one screen at a time (*Chong et al., 2015*; *Tkach et al., 2012*; *Kraus et al., 2017*). Although these approaches can discover differences in the abundance

and localization of proteins in response to perturbations, they have not been applied in integrative analyses: visual inspection is not scalable, and supervised machine learning usually requires at least some retraining for application to new datasets (although recent work endeavours to reduce this [*Kraus et al., 2017*]). On the other hand, unsupervised analysis of protein localization is expected to be scalable and does not require retraining. Indeed, unsupervised cluster analysis was applied to spatial mass spectrometry data following a single drug treatment to capture changes in relative distribution between the cytoplasm, nucleus, and nucleolus (*Boisvert et al., 2010*).

Recently, we developed an unsupervised localization change detection algorithm (*Lu and Moses, 2016*) that requires no training of parameters on each screen, easily scales to large image collections, and describes relative patterns of change quantitatively rather than relying on predefined classes, and is therefore applicable to all subcellular localization patterns. Here, we use this method to analyze datasets generated by the large-scale analysis of the yeast GFP collection (*Huh et al., 2003*). We exploit the scalability of this method to perform an integrative analysis that combines information about protein localization changes from multiple datasets. We reasoned that this analysis would provide contextual information that would allow us to determine whether a protein localization change is specific to a perturbation or might reflect a more general response, leading to a better understanding of how protein localization changes allow cells to adapt to a wide range of genetic and environmental situations. Using this context, we demonstrate that we can infer novel biology from our analysis.

We present, to our knowledge, the first quantitative integrative cluster analysis of protein localization changes. We show that our data acquisition and analysis methods can scale to analyses of hundreds of thousands of images. We find that protein localization changes are varied in their degree of specialization: some occur only in one perturbation tested, whereas others appear to be more general stress responses. Moreover, we find functionally specific themes for many clusters of protein localization change profiles, consistent with the known biological effects of the perturbations. We show that although some protein localization changes are accompanied by transcriptional or protein abundance changes, subcellular localization is in general an independent layer of regulation, and that shared patterns of localization changes cannot be explained simply by physical interactions or subcellular compartments.

## Results

### An unsupervised analysis of protein localization changes in more than 280,000 images

In this work, we sought to understand protein localization changes across a wide range of perturbations. To achieve this, we made use of datasets that are available in the CYCLoPs database (*Koh et al., 2015*): 15 image-based screens of the yeast GFP-fusion collection (*Huh et al., 2003*). These images show protein localization in untreated wild-type cells (three replicates), in an *rpd3Δ* deletion strain (three replicates), and three time-points each of wild-type cells subjected to rapamycin (RAP), hydroxyurea (HU), and α-factor (αF) treatment (*Chong et al., 2015*; *Kraus et al., 2017*). We also included data from two independent screens of the GFP-fusion collection in strains deleted for *IKI3*, a component of the elongator complex and a histone acetyltransferase that functions as part of the RNA polymerase II holoenzyme (*Wittschieben et al., 1999*). Altogether, this dataset encompasses 4143 individual yeast GFP-fusion proteins for a total of 281,724 images and a total of 15.5 million cells used.

*Figure 1* summarizes our method for representing protein localization changes across image screens. First, we start with two sets of micrographs of yeast cells, one of cells under untreated wild-type conditions and another under perturbation, where the protein of interest has been tagged with GFP, and where a cytosolic RFP has been expressed to facilitate cell identification. These images have been segmented into single cells (*Figure 1(i)*). We bin single cells into five bins by their stage in the cell cycle (*Figure 1(ii)*), and then by whether the cell is a mother or a bud cell (*Figure 1(iii)*), resulting in 10 independent bins in total; this binning strategy allows us to determine whether protein localization changes are dependent on cell cycle or type.

For each single cell, we measure a set of five interpretable features (*Figure 1[iv]*) that describe the average distribution of GFP-tagged proteins relative to certain cell landmarks such as the cell

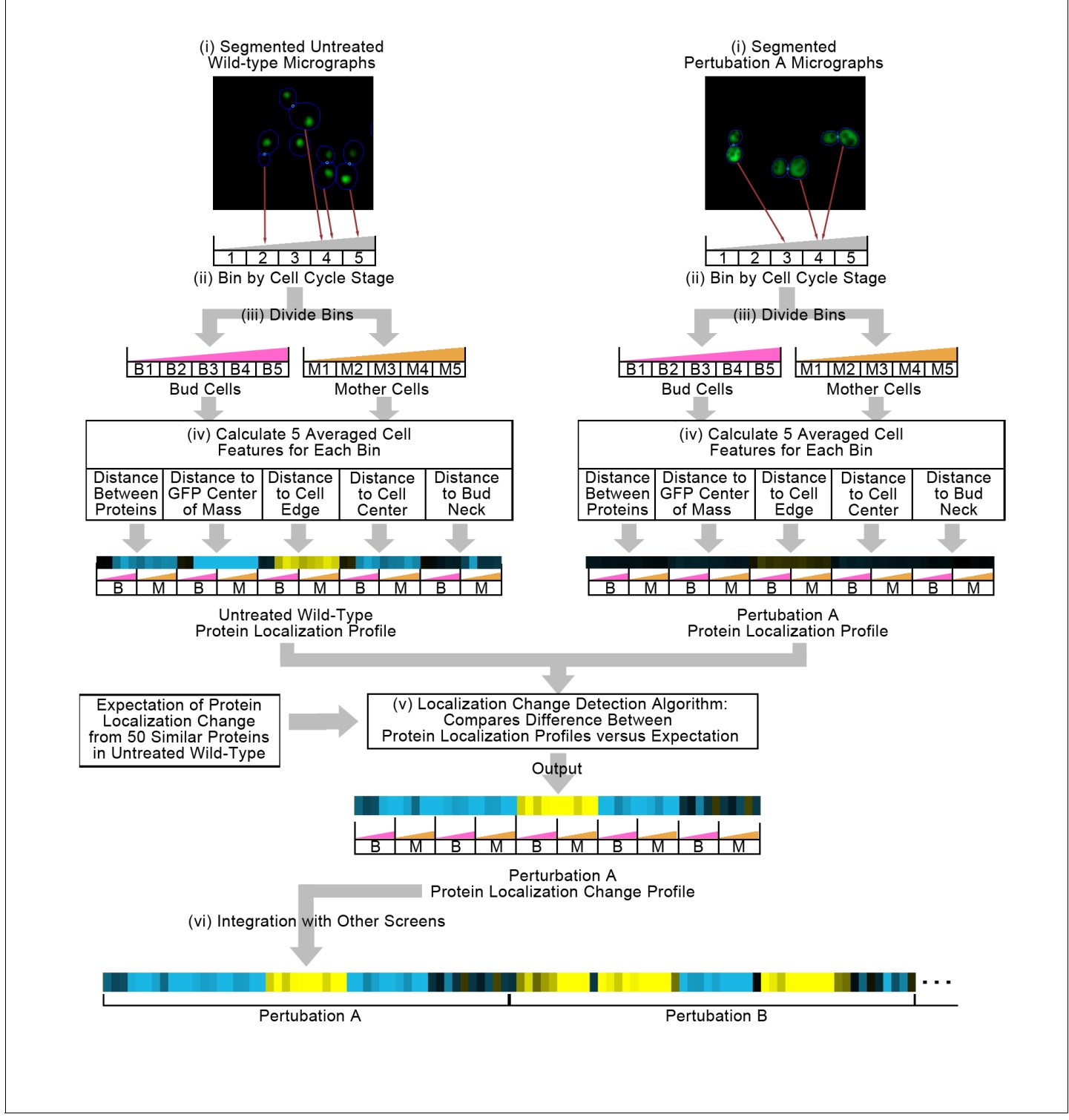

**Figure 1.** Summary of the methods used in this work. In step (i), we segment micrographs into single cells. We divide single cells into 10 bins in step (ii), first into five bins by cell cycle stage, and then by mother or bud cell type. We calculate five features for all single cells in step (iii), and report the truncated average of each bin for each feature to form the protein localization profile (see Materials and methods). To compare the untreated wild-type protein localization profile to the perturbation protein localization profile, we apply an unsupervised localization change detection algorithm in step (iv), which outputs a protein localization change profile comprised of a z-score for each feature in the protein localization profile, reporting how significant the difference between the wild-type and perturbation is relative to expectation for each feature. Finally, we concatenate different protein localization

*Figure 1 continued on next page*

*Figure 1 continued*

change profiles with each other in step (v) to integrate profiles from different image screens. Steps (i) to (iv) are previously published methods, and are described in the Materials and methods section. We represent profiles as heat maps in this work (see the *Figure 2* legend for details).

DOI: https://doi.org/10.7554/eLife.31872.003

The following figure supplement is available for figure 1:

**Figure supplement 1.** Quality control scatterplots comparing the protein localization change profile magnitudes for each protein (calculated using the sum of squares over the features of the profile) between replicates of the same perturbation, and between different perturbations.

DOI: https://doi.org/10.7554/eLife.31872.004

center or cell edge (*Handfield et al., 2013*). We average these measurements for single cells within each of the ten bins described above, resulting in a 50-feature vector that we term the 'protein localization profile'. As the name suggests, the protein localization profile represents the average localization of the GFP-tagged protein inside of a cell for a given image screen.

Previously, we have shown that these protein localization profiles are not directly comparable from image screen to image screens because of the presence of 'global effects', or systematic biases in image screens (*Lu and Moses, 2016*). To correct for this, we use an unsupervised localization change detection method (*Lu and Moses, 2016*) that compares the difference between a pair of protein localization profiles to a local expectation of the global effect for each protein (*Figure 1[v]*). The unsupervised localization change detection method converts a pair of protein localization profiles into a 'protein localization change profile' (or more simply a 'protein change profile'), which represents the expectation of protein localization change for a protein between two image screens. In this work, the protein localization change profile is a vector of 50 z-scores, each corresponding to a feature in the protein localization profile and describing the direction and magnitude by which each feature deviates from the expectation of there being no localization change.

We apply this method to all proteins and image screens. For the work that we present here, we compared all other image screen to an untreated wild-type screen that we designated as our reference untreated wild-type (labelled as WT2 in figures, corresponding to its database label in CYCLoPS [*Koh et al., 2015*]). For each protein, we concatenated protein localization change profiles to integrate results from different image screens (*Figure 1* [vi]). We found that the unsupervised localization change detection method predicts reproducible localization changes between replicate image screens: the magnitude of the protein change profile is strongly correlated between replicates of the same perturbation (*Figure 1—figure supplement 1A and B*), but weakly correlated between different perturbations (*Figure 1—figure supplement 1C and D*).

Finally, we grouped proteins by clustering our concatenated protein change profiles. The heat map in *Figure 2* shows clusters formed by the 1159 profiles with strong signals in at least three features. We found that the concatenated vectors are readily interpretable. Because the protein localization change profile is a relative measure, the profiles generally have large values for image screens in which the protein changes localization, and low values for those in which it does not.

We manually evaluated images for localization changes predicted by our protein change profiles for five distinct clusters and recorded the proportion of proteins with localization changes visible to a human evaluator (*Supplementary file 1*). In general, we observed that clusters that had strong signals in their protein change profiles had a higher ratio of visible localization changes; Clusters E and J in *Figure 2* have an average protein change profile magnitude of 33.5 and 34.5 for the perturbations that are predicted to change their localization, and contain 9/17 and 18/25 visible localization changes, respectively. Proteins that did not have visible localization changes within these clusters tended to have weaker signals (proteins without an asterisk in *Figure 2—figure supplement 1*). By contrast, clusters with more moderate signals in their protein change profiles contain fewer localization changes that are visible to the human eye, or more ambiguous or borderline protein localization changes. Cluster F in *Figure 2* has an average protein change profile magnitude of 14.3, and contains 2/19 visible localization changes, with a further 9/19 localization changes being too ambiguous to determine. The localization changes within this cluster were visually subtle, involving changes from a round nuclear phenotype in the untreated wild-type to an irregular 'fluffy' nuclear phenotype in the perturbation (Tma16 and Ysh1 in *Figure 2—figure supplement 2B*).

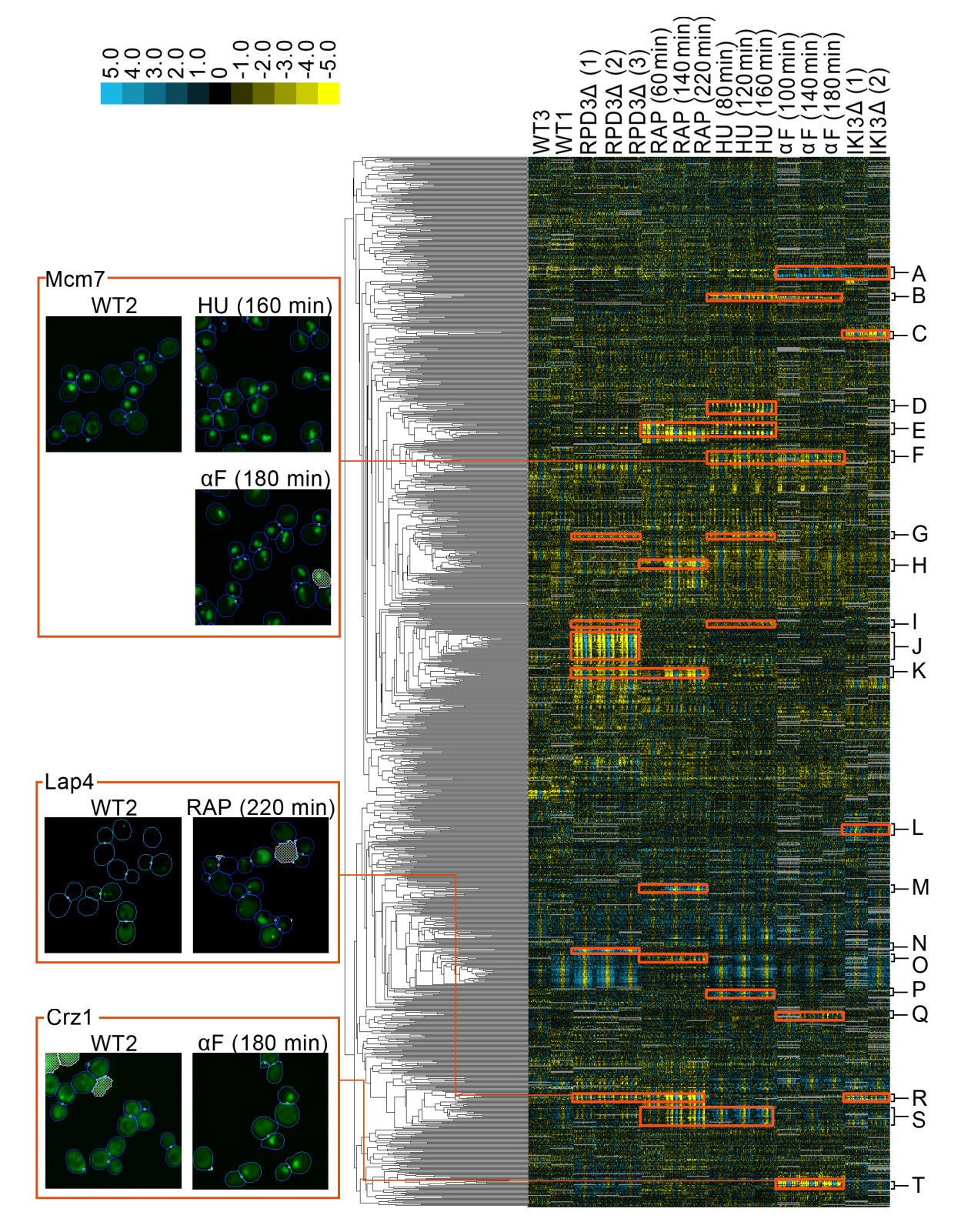

**Figure 2.** A clustered heat map of protein change profiles. The right panel shows the 1159 protein change profiles with the strongest values, grouped by similarity. In this visualization, rows represent the protein change profiles for individual proteins, whereas columns represent feature measurements, with each set of 50 measurements corresponding to an image screen. Positive values are blue, whereas negative values are yellow, with the intensity of color corresponding to the magnitude of the value, as shown in the color bar. Grey values are missing data. The headers indicate the image screens

*Figure 2 continued on next page*

*Figure 2 continued*

that sets of features correspond to, abbreviated using WT to indicate the untreated wild-type screens, ΔRPD3 for the *rpd3Δ* replicates, RAP for time points of the rapamycin treatment, HU for time points of the hydroxyurea treatment, αF for time points of the α-factor treatment, and ΔIKI for the *iki3Δ* replicates. The dendrogram depth shows similarity between connected protein profiles or groups of profiles. We highlight examples of strong patterns of protein change profiles in yellow, with some clusters that we have annotations for labelled from A to T, with labels and enrichments for some clusters presented in *Table 1*. In the four boxes on the left, we show examples of localization changes found in our clusters of protein change profiles. The images are representative cropped micrographs of yeast cells, where the protein named in the top left corner of each box has been tagged with GFP (shown as the green channel). The blue lines in the images show the boundaries drawn between cells by our single-cell segmentation algorithm, the small white circles between cells indicate mother-bud relations, and the white meshed regions indicate areas that have been ignored by our image analysis because they are likely to be artifacts or mis-segmented cells.

DOI: https://doi.org/10.7554/eLife.31872.005

The following source data and figure supplements are available for figure 2:

**Source data 1.** Protein localization change profiles for all of the perturbations presented in *Figure 2*.
DOI: https://doi.org/10.7554/eLife.31872.009

**Figure supplement 1.** Heat maps comparing the protein localization change profile with the transcript change and protein abundance change for three clusters from *Figure 2* (see legend of *Figure 2* for details on the heat map visualization).
DOI: https://doi.org/10.7554/eLife.31872.006

**Figure supplement 2.** (A) Heat map of the protein localization change profiles for Cluster F and (B) cropped representative micrographs for some proteins in Cluster F from *Figure 2* (see legend for *Figure 2* for details on the heat map visualization.
DOI: https://doi.org/10.7554/eLife.31872.007

**Figure supplement 3.** (A) Heat map of the protein localization change profiles for Cluster Q shown in *Figure 2*, and (B) cropped representative micrographs for some proteins in Cluster Q.
DOI: https://doi.org/10.7554/eLife.31872.008

## Clustering of multi-perturbation z-score vectors reveals shared and specific localization changes that are supported by the literature

To investigate the hypothesis that our clusters of protein localization change profiles contain proteins that are linked functionally, we evaluated clusters that showed strong signals for perturbations. We ignored clusters that had strong signals in the other untreated wild-type screens (WT3 and WT1 in *Figure 2*), as these proteins were predicted to change localization between untreated wild-type screens. These localization changes can occur for a variety of reasons, including inherent protein variability (*Handfield et al., 2015*), day-to-day variation, or strain-specific variation; because localization changes for these proteins may not be specifically caused by the perturbation, we exclude them from our analysis. *Table 1* shows GO ontology enrichments for clusters from *Figure 2*. In this section, we provide examples in which our clustering reveals known biology.

We reasoned that the rapamycin and hydroxyurea treatment profiles would share localization changes, because both drugs induce yeast stress responses (*Loewith and Hall, 2011*; *Koç et al., 2004*). We found a cluster of protein change profiles (*Figure 2E*, *Table 1E*) that predicted localization changes in both drug treatments. Within this cluster, we found stress-responsive proteins such as Msn2, a master regulator of the yeast general environmental stress response (*Gasch et al., 2000*). In addition, this cluster also contained proteins that are involved in ribosomal biogenesis and rRNA processing, consistent with previous work showing repression of ribosomal subunit gene transcription following the environmental stress response (*Gasch et al., 2000*). We observed some subtle differences in the protein change profiles of particular proteins in this cluster; for example, while the relocalization of Msn2 was shared between the rapamycin and hydroxyurea perturbations, the relocalization of Tsr1, a protein predicted to move from the cytoplasm to the nucleolus following stress (*Lee et al., 2014*), was shared between the *rpd3Δ*, rapamycin, and hydroxyurea perturbations. In another example, Msn2 was predicted to differ in localization relative to the reference untreated wild-type in all three time points of hydroxyurea treatment but in only the first time point following growth in the presence of rapamycin. By contrast, Stb3, a repressor of growth in response to stress that is known to be regulated by relocalization between the nucleus and cytoplasm (*Liko et al., 2010*), was predicted to differ in all time points of both perturbations.

Not all stress-related proteins clustered together; for example, Lap4, a hydrolase that relocates from the cytoplasm to the vacuole under starvation conditions (*Suzuki et al., 2002*), was predicted to relocalize in the rapamycin and *rpd3Δ* perturbations, but not in the hydroxyurea perturbation. For

**Table 1.** Annotations for some labelled clusters presented in *Figure 1*.
The letter in the cluster column corresponds to the letters assigned to clusters in *Figure 2*. Selected significant gene ontology enrichments are listed with their respective GO accession numbers and p-values, as well as some arbitrarily chosen examples of proteins. Clusters that are not associated with a GO accession number are manual annotations. Under the 'Proteins with annotation' column, we report the number of proteins with the GO annotation versus the total number of proteins in that cluster.

| Cluster | Perturbation(s) | Annotation | P-value | Proteins with annotation | Examples |
|---|---|---|---|---|---|
| A | α-factor, iki3Δ | Cytoskeleton-dependent cytokinesis [GO:0061640] | 3.77E–07 | 8/12 | Bud3, Bud4, Cdc10, Cdc11, Hof1, Myo1 |
| | | Cell cycle [GO:0007049] | 9.21E–05 | 11/12 | |
| D | Hydroxyurea | Cell cycle | – | – | Cdc28, Cdc7, Clb2 |
| E | Rapamycin, hydroxyurea | RNA metabolic process [GO:0016070] | 6.70E–04 | 15/18 | Msn2, Dot6, Rtg1, Rtg3, Enp1, Tsr1, Stb3, Utp20 |
| | | Stress-responsive transcription factors | – | – | Msn2, Dot6, Rtg1, Rtg3 |
| F | Hydroxyurea, α-factor | Extrinsic component of vacuolar membrane [GO:0000306] | 0.028823 | 3/19 | Iml1, Vac14, Tco89 |
| | | DNA replication | – | – | Pol12, Mcm7, Ctf18 |
| G | rpd3Δ, hydroxyurea | Triglyceride lipase activity [GO:0004806] | 0.029239 | 2/6 | Tgl4, Tgl5 |
| | | Metabolic activity | – | – | Tgl4, Tgl5, Rnr3, Gdb1 |
| H | Rapamycin | Negative regulator of hydrolase activity [GO:0051346] | 2.70E-04 | 4/9 | Stf1, Inh1, Yhr138c, Pbi2 |
| | | Enzyme inhibitor activity [GO:0004857] | 0.001441 | 4/9 | |
| J | rpd3Δ | Annotated by *Chong et al. (2015)* to localize away from cytoplasm in rpd3Δ | – | – | Acs1, Rad7, Mni1, Yjr008w, Ycr061w, Pab1, Pus4, Yor292c |
| L | iki3Δ | Ribosomal subunits | - | - | Rps9a, Rpl13b, Rps0b |
| M | Rapamycin | Cytosolic ribosome [GO:0022626] | 0.010879 | 4/5 | Rpl23a, Rpl40a, Rpl40b, Rps10a |
| | | Ribosomal subunits | – | – | |
| Q | α-factor | Golgi vesicle transport | 0.005435 | 5/7 | Apl2, Avl9, Gyl1, Bch1, Gyp5 |
| | | Cellular bud tip | 9.28E–06 | 5/7 | Rgd2, Yhr182w, Avl9 Gyl1, Gyp5 |
| R | rpd3Δ, rapamycin, iki3Δ | Vacuole [GO:0005773] | 9.39E–05 | 7/7 | Bap2, Itr1, Hxt2, Aqr1, Dip5 |
| | | Ion transmembrane transporter activity [GO:0015075] | 0.020585 | 5/7 | |
| S | Rapamycin, hydroxyurea | Bounding membrane of organelle [GO:0098588] | 0.027761 | 8/12 | Ams1, Kch1, Ymd8, Pho91, Mnn2, Och1, Kre2, Gnt1 |
| | | Protein glycosylation [GO:0006486] | 0.020786 | 4/12 | Mnn2, Och1, Kre2, Gnt1 |
| T | α-factor | Cellular response to pheromone [GO:0071444] | 5.37E–04 | 5/9 | Fus1, Kar4, Ste2, Aga2, Crz1 |

DOI: https://doi.org/10.7554/eLife.31872.010

The following source data is available for Table 1:
Source data 1. A spreadsheet containing lists of the proteins in each cluster highlighted in *Figure 2*, and the GO enrichments of these clusters.
For GO enrichments, we report the enriched term, the p-value, all proteins in the cluster annotated with the term, and the GO accession number.
DOI: https://doi.org/10.7554/eLife.31872.011

this reason, Lap4 clustered with other proteins that were predicted to relocalize in the rapamycin and *rpd3Δ* perturbations instead (*Figure 2R*, *Table 1R*). Our profile analysis of Lap4 is consistent with the image data. In the untreated wild-type, the protein is weakly cytoplasmic, with a small fraction of cells showing the protein in cytoplasmic foci. Under rapamycin and *rpd3Δ* perturbation, the protein is localized to the vacuole with a visually noticeable increase in the frequency of cells that have cytoplasmic foci (see Lap4 images in *Figure 1*).

We also found that sets of cell-cycle proteins could be distinguished by their protein change profiles. Treatment of yeast cells with hydroxyurea or α-factor induces cell cycle arrest during the S- and

G1-phases, respectively (*Day et al., 2004*; *Udden and Finkelstein, 1978*). We found that some proteins involved in DNA replication (Pol12, Mcm7) and in the regulation of transcription at the G1/S transition (Swi6) had strong patterns of signals in their protein change profiles for both the hydroxyurea and α-factor perturbations (*Figure 2F*, *Table 1F*). The Mcm complex, including Mcm7 which we show as an example in *Figure 1*, is known to relocalize according to cell-cycle stage (*Nguyen et al., 2000*). Consistent with this prior knowledge and the predictions made by our protein change profiles, images of Mcm7 in the untreated wild-type cells showed a heterogeneous mixture of cytoplasmic or nuclear-localized protein. In both hydroxyurea and α-factor perturbations, the localization was almost entirely nuclear, with the distribution of the nuclear-localized protein becoming irregular compared to the circular untreated wild-type nuclear distribution. A different cluster of proteins with strong patterns of signals for the α-factor perturbation also includes cell cycle proteins (*Figure 2A*, *Table 1A*); instead of being shared with the hydroxyurea perturbations, some of these localization changes are predicted to occur under *iki3Δ* perturbation too, consistent with the role of *iki3Δ* in regulating sensitivity to G1 cell-cycle arrest (*Butler et al., 1994*).

In addition to clusters that have patterns that are shared between multiple perturbations, we also found clusters with patterns of strong signals that are specific to only one perturbation. α-factor is a pheromone that induces the yeast mating response. We find a cluster of proteins involved in the mating response (*Figure 2T*, *Table 1T*), all of which have a protein change profile that predicts a relocalization specific to α-factor treatment, such as changes in the localization of the transcription factor Kar4, which is induced as a regulator of downstream events in the mating response (*Lahav et al., 2007*), and of the transcription factor Crz1, which has been shown to relocate from the cytoplasm to the nucleus during bursts of calcium induced by the mating pheromone (*Carbó et al., 2017*). Images for Crz1 in the untreated wild-type versus the α-factor screens (shown in *Figure 1*) confirm that although the protein is predominantly cytoplasmic in the untreated wild-type, a proportion of the cells under α-factor perturbation have Crz1 that is also localized to the nucleus. As another example of a cluster with profiles that predict relocalizations under one perturbation only, we found a cluster specific to rapamycin perturbation that includes proteins that regulate hydrolase activity (*Figure 2H*; *Table 1H*), which may relate to the autophagic processes induced by rapamycin (*Alvers et al., 2009*).

A list of proteins that change for each perturbation is available in *Supplementary file 2*, curated by selecting clusters of protein change profiles that have strong coordinated signals from the heat map in *Figure 2*.

## Clustering of protein localization changes reveals new patterns of protein relocalization and enables development of hypotheses about protein function

The extent to which we can interpret our clusters within known information from the literature varies. For example, while the mating response is well-studied and provides a clear basis to validate localization changes predicted by our method, other clusters of predictions were relatively novel. By integrating the observations made possible by our exploratory analysis of protein change profiles with a closer qualitative analysis of images of the proteins implicated in these clusters, the data can be mined to develop informed hypotheses for experimental follow-up. In this section, we describe three examples of this process.

### Stress-responsive transcription factors that exhibit stochastic pulsatility may be controlled by the same mechanism

We observed a small cluster of three transcription factors, Msn2, Dot6, and Rtg3, with patterns shared between the hydroxyurea and rapamycin perturbations. All three proteins exhibited a protein change profile that indicates that they were becoming more compact in distribution and closer to the cell center following both hydroxyurea and rapamycin treatment (*Figure 3A*). Evaluating the images (*Figure 3B*) confirmed that these proteins moved from the cytoplasm to the nucleus following these perturbations. The import of Msn2, Dot6, and Rtg3 into the nucleus is expected, as previous work has shown that all three move to the nucleus following inhibition of TORC1 kinase by rapamycin treatment (*Loewith and Hall, 2011*). Furthermore, Msn2 and Dot6 were found to localize

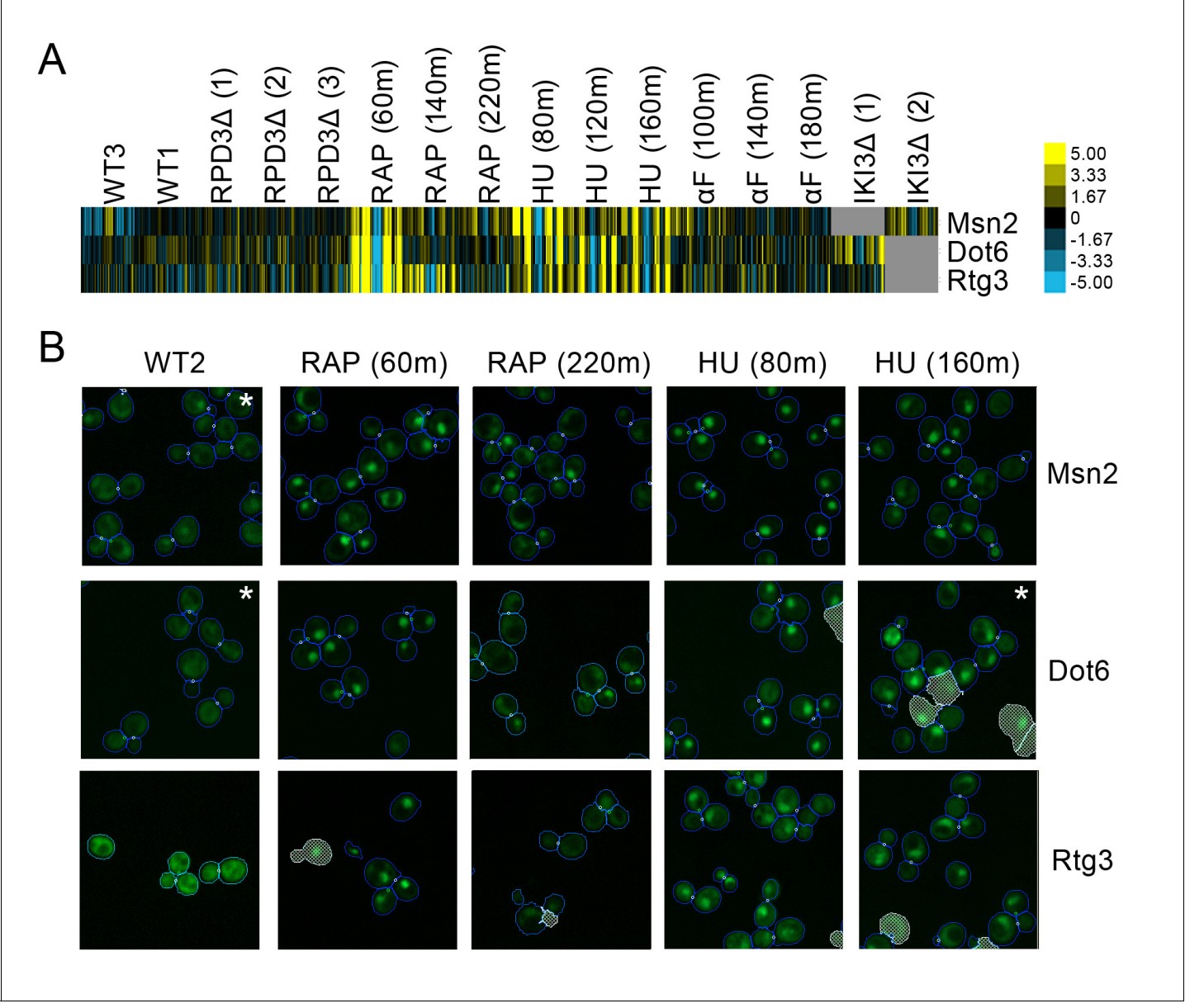

**Figure 3.** (A) Heat map of the protein change profiles and (B) cropped micrographs of the GFP collections strains for stress-responsive transcription factors Msn2, Dot6, and Rtg3. Refer to *Figure 2* for information on the heat map visualization of the protein change profiles and micrograph images. In (B), we show cropped micrographs of yeast cells in which Msn2, Dot6, and Rtg3 have been fused with GFP (green), under standard medium (WT2), rapamycin treatment for 60 min (RAP 60m) and 220 min (RAP 220m), and hydroxyurea treatment for 80 min (HU 80m) and 160 min (HU 160m). Cropped micrographs labelled with an asterisk (*) have had their image intensities clipped at their lower range to remove noise.

DOI: https://doi.org/10.7554/eLife.31872.012

to the nucleus in previous manual assessment of images following hydroxyurea exposure (*Tkach et al., 2012*).

We detected interesting coordinated time dependencies in these responses. All three transcription factors were localized to the nucleus throughout all three time points of hydroxyurea perturbation, but strongly localized to the nucleus at the first time point of rapamycin perturbation before returning closer to the phenotype for untreated cells in later time points. Although there were subtle differences between the localization patterns of the three proteins (for example, Msn2 only diminishes in relative localization to the nucleus rather than returning fully to the cytoplasm as we see in the other proteins), the general trend was preserved.

These similarities in their protein change profile patterns led us to hypothesize that Msn2, Dot6, and Rtg3 may be regulated by a common mechanism. Msn2 is the most studied of the three transcription factors, and exhibits a stochastic oscillation (or pulsing) between the cytoplasm and nucleus that is regulated by the cAMP-protein Kinase A (PKA) pathway (*Jacquet et al., 2003*). Moreover, Msn2 pulsing responds to stress in a quantitative manner, with low stress levels inducing a cytoplasmic steady state, high stress inducing a nuclear steady state, and only intermediate levels of stress inducing pulsing (*Jacquet et al., 2003*); rapamycin treatment specifically induces nuclear accumulation of Msn2 (*Beck and Hall, 1999*). Proteome-wide screens have revealed comparable dynamics for Dot6; whereas Rtg3 was not found to have pulsatile dynamics (*Dalal et al., 2014*).

Given our discovery that Rtg3 shares a protein change profile with Dot6 and Msn2, we decided to revisit the condition-specific dynamics of Rtg3 nuclear localization. To test whether Rtg3 also shows condition-specific pulsing dynamics, we produced time-lapse movies of Rtg3 in standard growth media and in medium containing rapamycin.

We observed Rtg3-GFP protein pulses (in both the GFP-collection strain and in an independently constructed Rtg3-GFP fusion strain, see Materials and methods) during growth in standard medium. We found that, as predicted by our cluster analysis, rapamycin treatment increased the duration of Rtg3 accumulation in the nucleus. In standard medium, single cells showed frequent oscillation of Rtg3 between the nucleus and cytoplasm, while under rapamycin treatment, single cells typically showed a prolonged period of nuclear localization in our movies. Quantification of the dynamics of Rtg3-GFP pulsing (*Figure 4*) revealed that the average period of nuclear localization for rapamycin-treated cells (175 min) was significantly greater (p=0.0003, single-tailed t-test) than the average for untreated cells (123 min). We repeated this experiment with a lower exposure time (400 ms rather 700 ms) and similarly found that the average period of nuclear localization for rapamycin-treated cells (216 min, n = 53) was also significantly greater (p=0.0103, single-tailed t-test) than the average for untreated cells (168 min, n = 41).

## Sets of ribosomal subunits exhibit perturbation-specific responses

We found evidence of localization changes for many ribosomal subunits, with their protein change profiles predicting that subsets of ribosomal proteins may respond specifically to one perturbation. As an example, we show a set of ribosomal protein subunits in *Figure 5* that had a pattern of strong signals in its protein change profile in the last two timepoints following rapamycin treatment, with some proteins showing a redistribution from the nucleus to the cytoplasm in the images. These perturbation-specific localization changes for particular ribosomal protein subunits contrast with the more general transcriptional response to stress of ribosomal protein genes (*Gasch et al., 2000*).

A growing body of research suggests that specific ribosomal subunits may have extra-ribosomal functions ranging from ribosomal biogenesis to DNA repair and adhesive growth (*Lu et al., 2015*). For example, Rpl40a and Rpl40b are ubiquitin-ribosomal protein fusions that contribute to Rps27b pre-rRNA maturation in the nucleolus by the cleaving of the ubiquitin-fusion precursor and to Rps60 maturation by export of the protein into the cytoplasm (*Fernández-Pevida et al., 2012*). We found that Rpl40a and Rpl40b both redistributed from the nucleus to the cytoplasm following prolonged rapamycin treatment (Rpl40a is shown as an example in *Figure 4b*), which may reflect the reduction in ribosomal biogenesis and rRNA synthesis and the processing activity caused by rapamycin (*Stauffer and Powers, 2015*).

## Membrane proteins exhibit a reciprocal pattern of localization changes following hydroxyurea or rapamycin treatment

We observed a cluster of proteins that had strong protein change profile patterns in both the rapamycin and hydroxyurea perturbations, but with different patterns in each perturbation. Following rapamycin treatment, the direction of localization changes indicated that the proteins were becoming closer to the cell center and further from the cell edge relative to those in the untreated cell, whereas in following hydroxyurea treatment, the same proteins were predicted to relocalize further from the cell center and closer to the cell edge. Evaluation of the images of cells expressing these proteins (*Figure 6*) showed that most were either plasma-membrane- or Golgi-localized proteins. In untreated cells, these proteins were localized to both the membrane or Golgi and the vacuole; following hydroxyurea treatment, the distribution of protein shifts towards the membrane, whereas

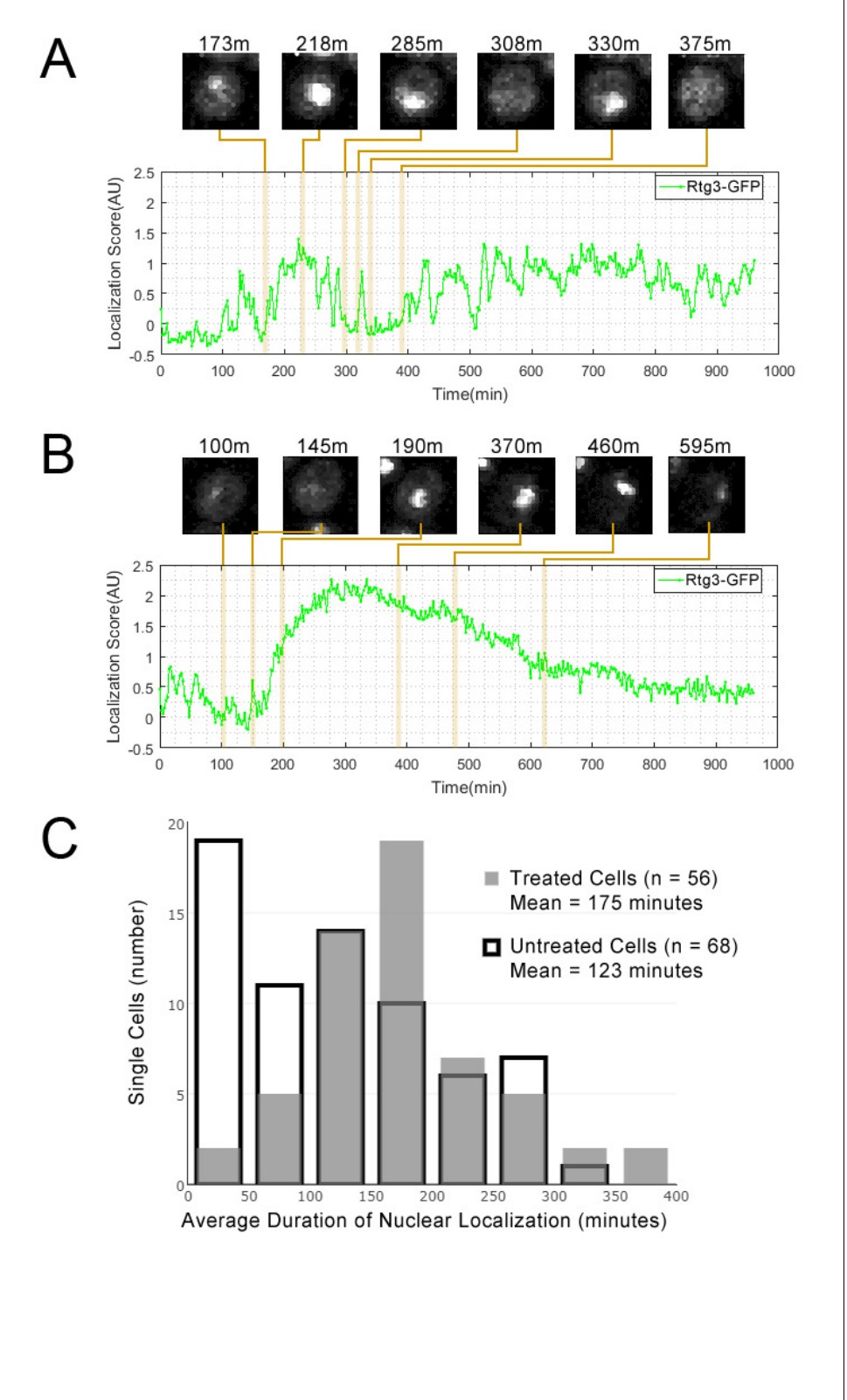

**Figure 4.** Single cell localization dynamics for Rtg3 for (**A**) a cell with frequent oscillations between the nucleus and cytoplasm, representative of typical untreated cells, and for (**B**) a cell with a single prolonged pulse, representative of typical rapamycin-treated cells. (**C**) A histogram of the average duration of nuclear localization of Rtg3 for single cells treated with rapamycin versus untreated cells. For rapamycin-treated cells, rapamycin is added at 0 min. To quantify the localization of Rtg3-GFP in single cells, we compute a localization score (expected nuclear intensity, see Materials and methods),

*Figure 4 continued on next page*

*Figure 4 continued*

which is higher when there is more nuclear-localized protein and lower when there is more cytoplasmic protein. We track these scores for each single cell over each frame of our movie to track the cells over time. Plots show the localization score over time. We show cropped stills from the time-lapse movie showing the state of the cell at various timepoints (labels show the timepoint in minutes).

DOI: https://doi.org/10.7554/eLife.31872.013

following rapamycin treatment, the distribution shifted towards the vacuole, confirming the prediction of our protein change profiles.

## Cluster associations found by our unsupervised exploration of localization changes are complementary to other high-throughput experiments

Next, we explored the relationship between our protein change profile clusters and other high-throughput datasets.

First, we compared the associations between proteins found in our protein localization change analysis to those found in transcript and protein abundance data, by comparing the protein profile similarities in each data source. We expect our clusters to overlap partially with the clusters retrieved from these other datasets, such as in cases where protein localization changes are being driven by or forming a feedback loop with transcript and abundance changes, but we anticipated that at least some protein localization changes are an independent layer of regulation. We constructed profiles for transcript changes and protein abundance changes for the perturbations that

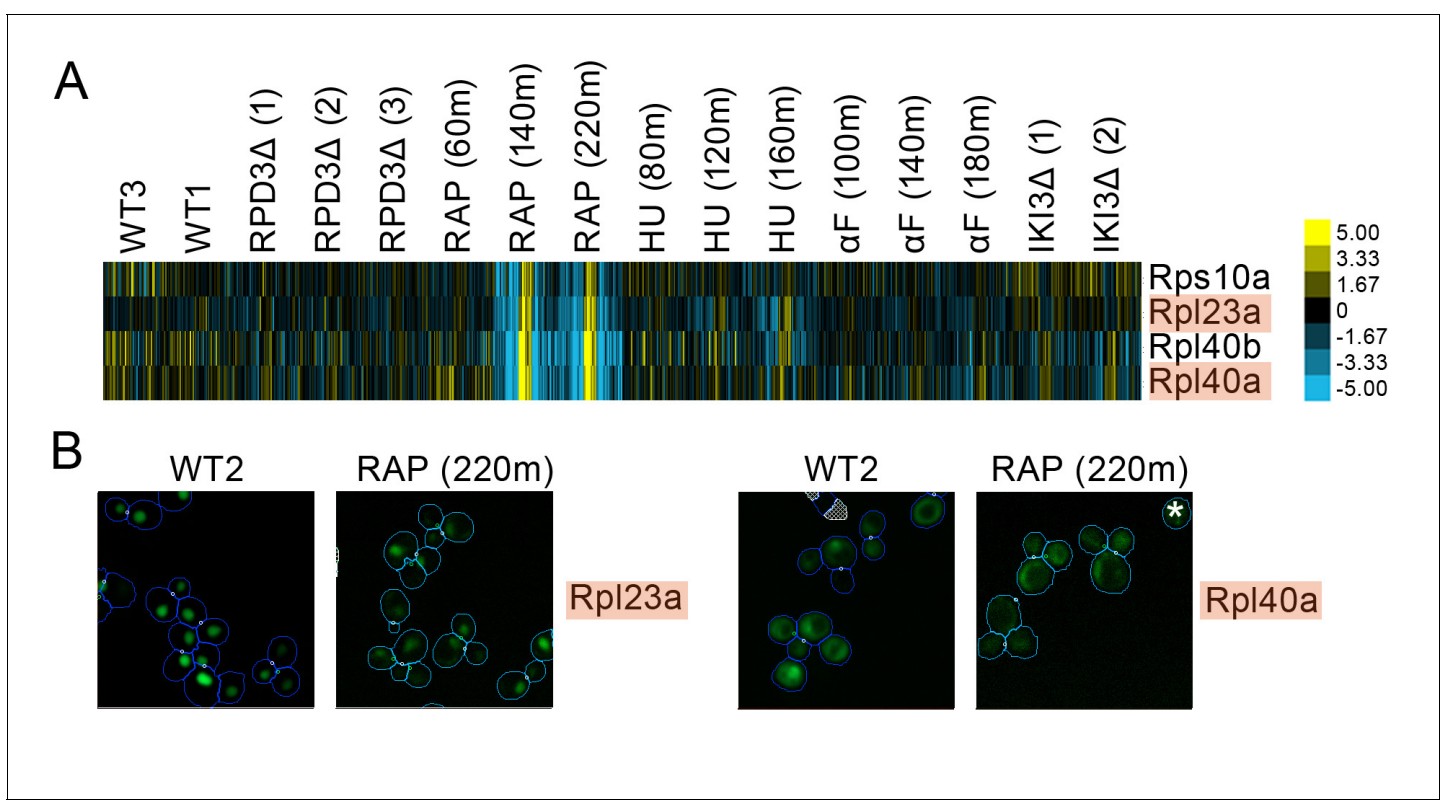

**Figure 5.** (A) Heat map showing protein change profiles, and (B) cropped micrographs for ribosomal subunits belonging to a cluster that responded specifically in rapamycin. Refer to *Figure 2* for information on the heat map representation of the protein change profiles and micrograph images. In (B), we show cropped micrographs where the labelled protein has been respectively fused with GFP (green), in standard media (WT2) and after 220 min of rapamycin treatment for the proteins highlighted in blue in (A). Cropped micrographs labelled with an asterisk (*) have had their image intensities clipped at their lower range to remove noise.

DOI: https://doi.org/10.7554/eLife.31872.014

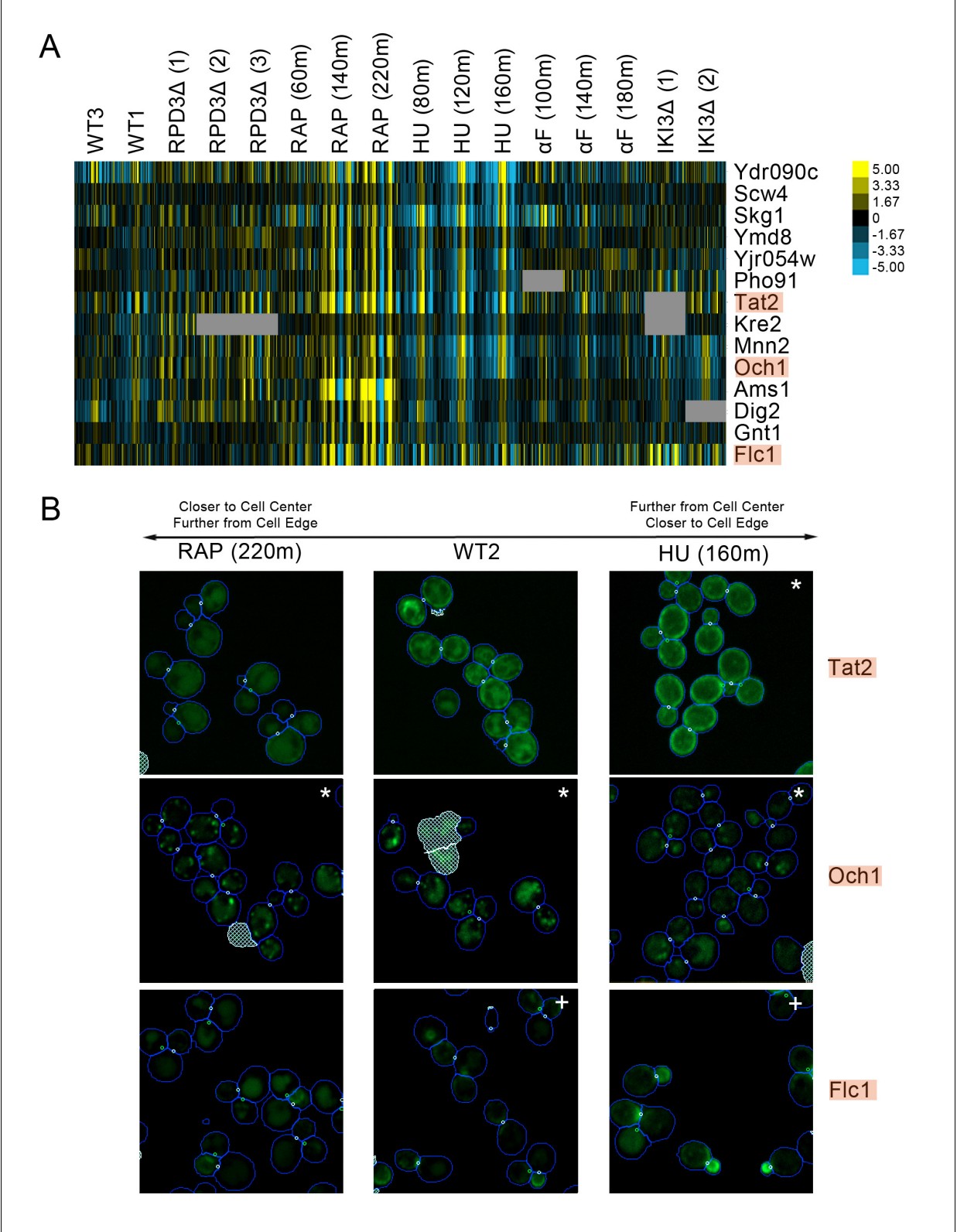

**Figure 6.** (A) Heat map of the protein change profiles and (B) cropped micrographs for a cluster of proteins with reciprocal protein change profiles between hydroxyurea and rapamycin treatment. Refer to *Figure 2* for information on the heat map representation of the protein change profiles and micrograph images. In (B), we show cropped micrographs of yeast cells in which Tat2, Och1, and Flc1 (highlighted in blue in [A]) have been tagged with GFP (green), after 220 min of rapamycin treatment (RAP 220m), growth in standard media (WT2), and after 160 min of hydroxyurea treatment (HU

*Figure 6 continued on next page*

*Figure 6 continued*

160m). Cropped micrographs labelled with an asterisk (*) or a plus sign (+) have had their image intensities clipped at their lower range or higher range, respectively, to remove noise and to better visualize protein localization.

DOI: https://doi.org/10.7554/eLife.31872.015

we tested in *Figure 2*, and compared the pairwise distance between proteins for these profiles with those for our protein localization change profiles. *Figure 7A* and *Figure 7B* show scatterplots of the transcript and protein abundance profilesimilarities, respectively, against the protein localization change profile similarity.

We found little correlation between protein localization change profile similarity and transcript profile similarity (*Figure 7A*). Some clusters had more proteins with relatively small changes (low distances) in their transcription change profiles. Cluster T, which is enriched in proteins involved in the pheromone response (*Table 1*), had several protein–protein pairs with low localization change profile and transcript change profile distances (orange circles in *Figure 7A*), possibly due to the induction of these genes or proteins as a specific response to α-factor treatment. By contrast, Cluster S, which contains membrane and Golgi proteins with reciprocal localization changes, had higher transcription change profile distances for all pairs of proteins (purple circles in *Figure 7A*), suggesting that the concerted protein localization change behavior of these proteins may not occur at the transcriptional level. Finally, because we had previously detected unusual nuclear localizations for ribosomal subunits under specific perturbations, we decided to evaluate specifically the ribosomal subunits in this comparison (blue points in *Figure 7A*). In general, most pairs of ribosomal subunits have similar transcript change profiles, consistent with the transcriptional downregulation of ribosomal subunits globally as part of the environmental stress response (*Gasch et al., 2000*). By contrast, very few pairs of ribosomal subunits have low protein localization change profile distances, supporting our suggestion that localization changes in ribosomal subunits may be a more specific layer of regulation.

We found a greater correlation between protein localization change similarity and protein abundance change profile similarity, although some of this correlation may stem from the calculation of both profiles from the same dataset (see Materials and methods). Although many proteins that show similar localization change profiles have similar protein abundance change profiles, there are vast numbers of proteins with similar abundance change profiles that do not show similar localization change profiles. This suggests that similarity in protein localization changes is much more specific than similarity in protein abundance. We found that two clusters of protein localization change profiles had low distances in their protein abundance change profiles relative to the distribution of protein abundance change profile distances: Cluster J, which contains proteins that change abundance specifically in the Δ*rpd3* perturbation, and Cluster E, which contains stress-response proteins (*Table 1*).

To investigate the co-occurrence of protein localization changes and protein abundance changes directly, we plotted the magnitude of the protein change profile against the average protein abundance change for various screens in *Figure 7—figure supplement 1*. For all screens, we observed that in general, the strongest magnitude protein localization changes had low protein abundance changes; however, the Δ*rpd3* screen has a higher ratio of proteins with both large protein abundance and protein localization changes than the other perturbations. Finally, we directly compared the protein localization change profile with the transcript change and protein abundance change profiles for three clusters (*Figure 1—figure supplement 1*), and observed that except for Cluster J, most proteins with high protein localization change profile signals have weak transcription change and protein abundance change profiles. We note that the lack of correlation between the profiles cannot be explained by false positives alone, because the lack of correlation persists even for proteins for which we could manually evaluate a protein localization change (denoted with an asterisk (*) in *Figure 1—figure supplement 1*).

Next, we asked whether our clusters of protein localization change profiles were dominated by interacting proteins from the same complex. We hypothesized that the underlying causes behind localization changes were diverse, and that coordinated localization changes would not necessarily require that the participating proteins had to interact directly. To test overlap with physical protein–protein interactions, we evaluated the enrichment of our clusters of protein localization changes with

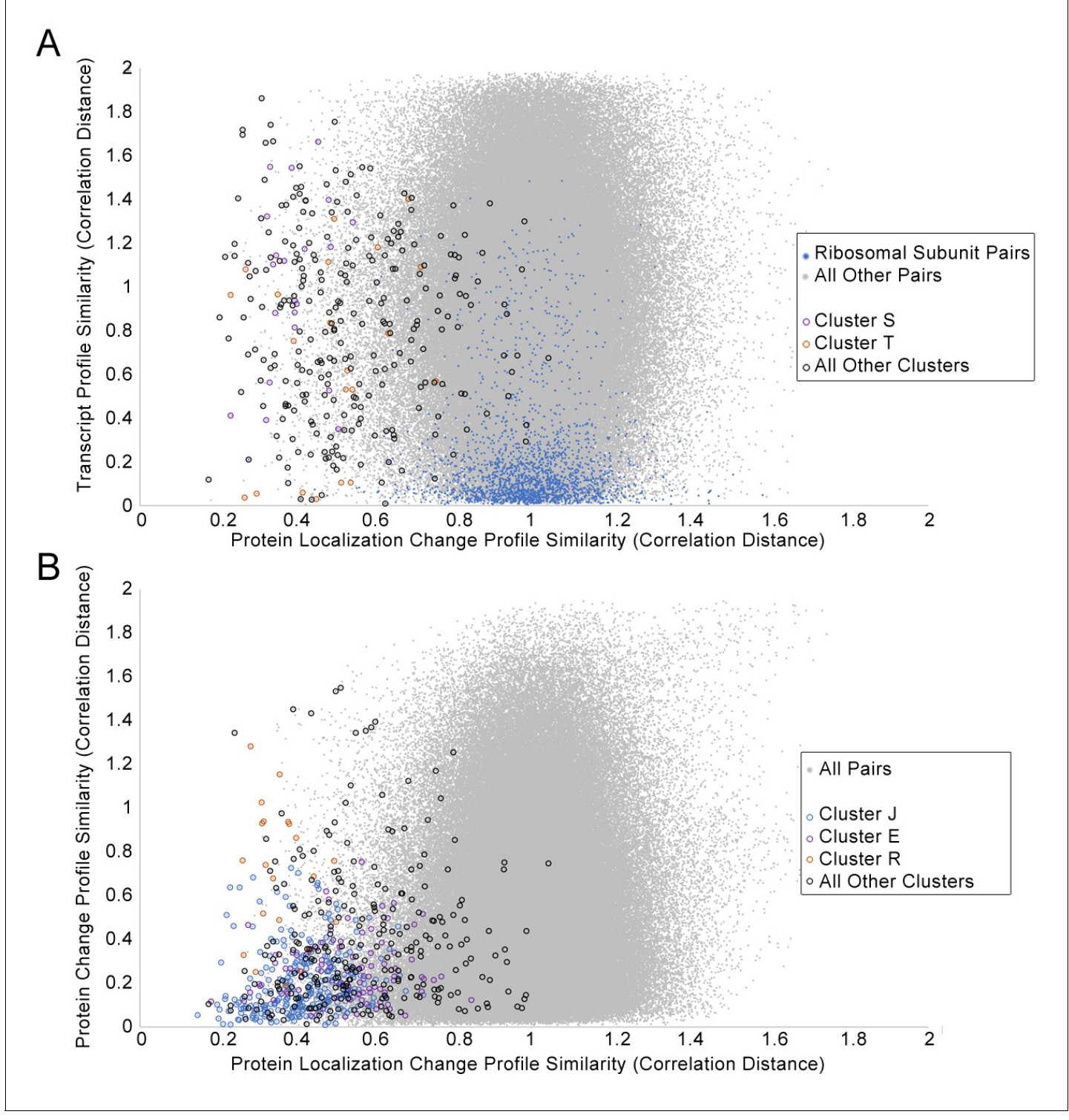

**Figure 7.** Scatterplots of the protein change profile similarity compared to (**A**) the transcript profile similarity and (**B**) the protein abundance profile similarity. We show protein–protein pairs between all proteins that had strong signals in either profile (see Materials and methods for details). Protein–protein pairs are shown as grey dots, and we circle all protein–protein pairs that belonged to a cluster identified in *Figure 2*. Some circled protein–protein pairs are color-coded depending on the cluster they belonged to, for some clusters of interest (refer to legend in figures). For the transcript profile similarity scatterplot in (**A**), we also visualize all protein–protein pairs between ribosomal subunits as blue dots.

DOI: https://doi.org/10.7554/eLife.31872.016

The following figure supplement is available for figure 7:

*Figure 7 continued on next page*

*Figure 7 continued*

**Figure supplement 1.** Scatterplots comparing the protein localization change profile magnitudes (calculated using the sum of squares over the features of the profile) with the average protein abundance change for each protein in four screens.

DOI: https://doi.org/10.7554/eLife.31872.017

high-quality physical interactions from the BIOGRID database (*Oughtred et al., 2016*), and all 'binding' protein associations in the STRING database (*Szklarczyk et al., 2017*) (*Figure 8A*). We observed that some clusters were enriched in protein–protein interactions in these databases: for example, Cluster A, which is enriched for cell-cycle proteins involved in cytokinesis (*Table 1*), was strongly enriched in both databases, and Cluster M, which contains ribosomal subunits, was strongly enriched in STRING. However, most clusters were not enriched, or only weakly enriched for protein–protein interactions.

Finally, we evaluated the degree to which proteins in our clusters of protein change profiles shared subcellular localizations in untreated wild-type cells. Previously, we observed a cluster of proteins in different compartments that exhibited similar behavior in response to perturbation: in the cluster of proteins that had strong protein change profile patterns in both the rapamycin and hydroxyurea perturbations, despite proteins being localized to the cell membrane and Golgi respectively, these proteins all move towards the vacuole under rapamycin treatment, and more to their specific localization under hydroxyurea treatment. These results suggested to us that our analysis was grouping proteins on the basis of shared trends in protein movement rather than simply grouping co-localized proteins moving from one compartment to another. To investigate the prevalence of clusters formed of proteins with different untreated wild-type localizations, we show the distribution of protein localization in the untreated wild-type for several clusters as pie graphs in *Figure 8B*, using manual annotations for localization classes for yeast proteins (*Huh et al., 2003*).

We find that clusters vary in their degree of heterogeneity in untreated wild-type localization. Some clusters are homogeneous: Cluster A, which was enriched for cell-cycle proteins, consists primarily of bud-neck localized proteins, and Cluster E, which included stress-responsive transcription factors (*Table 1*), consists primarily of proteins localized to the cytoplasm, nucleus, or cytoplasm or nucleus. As expected, Cluster S, which included the membrane and Golgi proteins that had strong protein change profile patterns in both the rapamycin and hydroxyurea perturbations, is highly heterogeneous. Cluster J, which contains proteins that change localization specifically in the Δ*rpd3* perturbation, is also highly heterogeneous. We also observed that Cluster T, enriched for proteins involved in pheromone response, consisted of an even mix of nuclear- and vacuole-localized proteins; this result reflects the biological process of mating, which includes both proteins that mediate the initial transcriptional response to pheromone, such as Kar4, as well as downstream factors such as Fus1, a cell fusion protein that facilitates vacuole mixing (*Nolan et al., 2006*), which is a later step in the mating process.

## Exploratory analysis of kinase deletion mutant screens identifies new patterns of localization change

Because the regulation of cellular processes by protein kinases is an area of major research focus in cell biology, we created a new set of image screens for seven kinase deletion mutants: *elm1Δ*, *hal5Δ*, *hsl1Δ*, *kin1Δ*, *kin2Δ*, *mck1Δ*, and *vhs1Δ*. Phosphorylation is a well-characterized mechanism for regulating protein localization (*Bauer et al., 2015*), so we anticipated that we would identify localization changes in these mutants. We found technical challenges in this dataset. First, the screens did not proliferate well, resulting in the inclusion of fewer cells (the average number of cells used for these screens was ~630,000, compared to an average of 1.1 million cells for the perturbed screens in our previous analysis), and the *elm1Δ* showed a dramatic morphologic change, such that cells grew in long chains (*Edgington et al., 1999*). Second, we did not collect replicates for these screens, making it more challenging to confirm the reproducibility of the signals that we observed. However, we were still able to find clusters containing proteins of interest, and we highlight two such clusters below. Data for all protein localization change profiles for the kinase deletion screens can be found in *Supplementary file 3*.

We found a cluster of proteins predicted to have localization changes for *elm1Δ* only (*Figure 9A*). This cluster included several stress-responsive proteins, such the proteins Apj1 and Pph21, both of

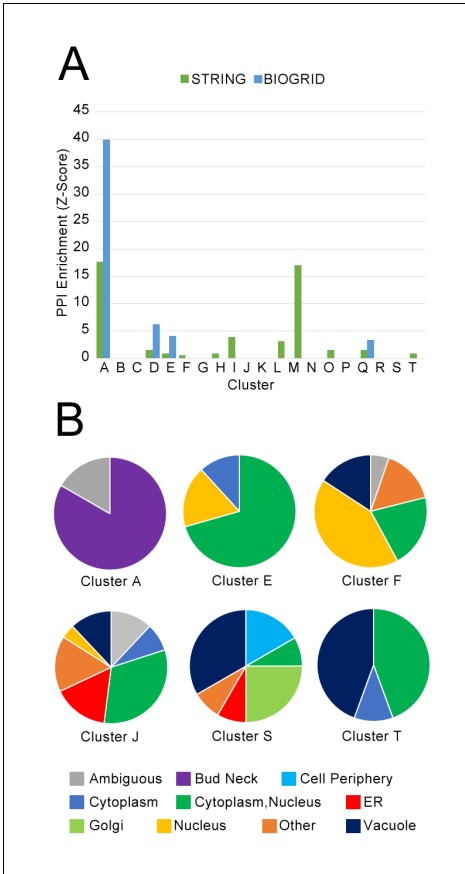

**Figure 8.** (A) Comparison of protein localization change profile clusters with physical protein–protein interactions and (B) localization in untreated wild-type cells. (A) The enrichment of each protein localization change profile cluster that we identified in *Table 1* for physical interactions in the BIOGRID and the STRING databases, measured as z-scores indicating degree of enrichment relative to null expectation. (B) The distribution of manually annotated subcellular localization for untreated wild-type proteins for proteins in Clusters A, E, F, J, S, and T.

DOI: https://doi.org/10.7554/eLife.31872.018

which have been found to form nuclear foci under DNA replication stress (*Tkach et al., 2012*). We found that a nuclear foci localization is observable for these proteins under *elm1Δ*, which is not seen in the wild-type (*Figure 9B*). Moreover, we also found that Gre3, which has been found to increase in abundance in response to DNA replication stress (*Tkach et al., 2012*), localized to the nucleus in *elm1Δ* relative to its cytoplasmic localization in the wild-type.

A cluster of proteins predicted to change localization predominantly in *hal5Δ* (*Figure 10A*) included some ribosomal subunits. Parallel to the observation described above in which we found that some ribosomal subunits showed specialized behavior in response to rapamycin treatment, we find that a different set of ribosomal subunits exhibit a stronger localization in the nucleus under *hal5Δ* perturbation relative to wild-type. Rps23a and Rpl7b are shown as examples in *Figure 9B*. We also observed that some nuclear transporters increased in their distribution to the nucleus specifically in the *hal5Δ* perturbation. Syo5 specifically facilitates the nuclear import of Rpl5 and Rpl11 (*Kressler et al., 2012*), whereas Nmd5 imports various proteins into the nucleus, including the transcription factors TFIIS (*Albertini et al., 1998*) and Crz1 (*Polizotto and Cyert, 2001*); both are shown as examples in *Figure 10B*.

Taken together, these results show that our integrative approach can be applied in a case in which there is little prior biological knowledge, and there are morphological differences between cells. This highlights the generalizability of the approach to a more realistic exploratory data analysis situation.

## Discussion

Here, we define quantitative associations between proteins on the basis of shared localization change behavior obtained through a systematic comparison of microscopy images, a data source that is infrequently exploited for integrative analysis. Although our study focuses on localization changes, images contain other information, including information about cell morphology (*Ohya et al., 2005*) and protein abundance (*Albert et al., 2014*). In principle, this information could also be incorporated into high-throughput integrative analysis.

This integrative approach contrasts with those used in previous studies that have produced lists of discrete protein localization changes that are readily interpretable by biologists, but not necessarily comparable or quantitative. For example, in our previous supervised approaches (*Chong et al., 2015*; *Kraus et al., 2017*), localization changes following individual perturbations were represented as flux networks, which show snapshots of protein movement between organelles following individual perturbations. However, it is not clear how the flux network obtained from one perturbation can be compared to that obtained from another perturbation. The value of integrating datasets is illustrated by the cluster of proteins involved in the mating response (Cluster T) and another cluster

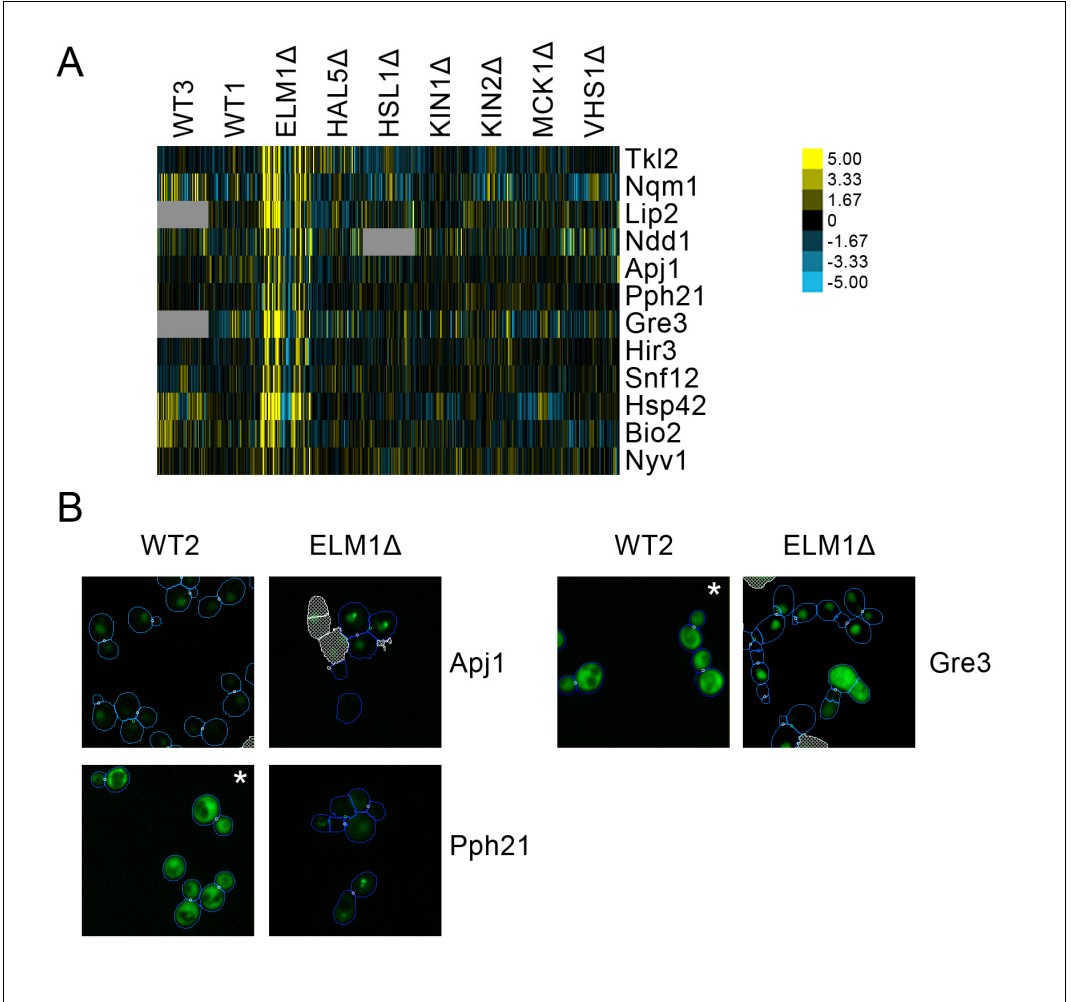

**Figure 9.** (A) Heat map showing protein change profiles, and (B) cropped micrographs for proteins in a cluster that responded specifically in *elm1Δ*. Refer to *Figure 2* for information on the heat map representation of the protein change profiles and micrograph images. In (B), we show cropped micrographs in which the labelled protein has been respectively fused with GFP (green), and the *elm1Δ* mutant for proteins Apj1, Gre3 and Pph21. Cropped micrographs labelled with an asterisk (*) have had their image intensities clipped at their lower range to remove noise.

DOI: https://doi.org/10.7554/eLife.31872.019

containing cell-cycle proteins (Cluster A). The former had localization changes in just α-factor, whereas the latter had changes in both α-factor and *iki3Δ* perturbations. Flux maps of the α-factor (*Kraus et al., 2017*) identify proteins from both sets; here, we separate these proteins into specific and functionally enriched groups. Thus, increased resolution offered by a multi-perturbation context empowers the identification of functional relationships between proteins.

Moreover, a comparative representation of protein localization changes between perturbations permits the detection of subtle differences in protein responses. We showed that regulators of the yeast stress response (such as Stb3 and Msn2) could be differentiated in subtle ways. Our work integrates multiple datasets as a compact visualization, allowing for a deeper understanding of the subtle conditional and time-dependent dynamics of proteins.

We addressed the question of how to scale analyses to tens of high-throughput microscopy screens. Previous approaches act on a single screen, scaling to tens of thousands of images. By integrating data from multiple screens, we show how to scale a single analysis to hundreds of thousands of images, an order of a magnitude increase in the scale of data.

Our approach detects a novel reciprocal pattern of localization changes for membrane proteins following hydroxyurea or rapamycin treatment. These protein responses are notable because they exhibit different localization changes depending upon the perturbation. Although we do not have a clear hypothesis for these changes, the observation was made clear by our visualization, which shows trends in localization changes simultaneously for both perturbations. That this pattern occurred in multiple membrane or Golgi-localized proteins, increases our confidence that it reflects important, yet currently unknown biology. Without clustering based on quantitatively comparable localization change profiles, the prevalence of the pattern would not have been apparent.

Our analysis is unbiased by prior biological knowledge. Despite this, we find many specific examples of known responses to perturbations, and thus strong evidence that the patterns we identify reflect underlying biology. The unbiased nature of our approach permits the association of new genes with previously known responses. We found that Crz1 was clustered with proteins in the mating response in the α-factor perturbation, a result that was initially unexpected given that Crz1 is not part of the canonical mating response (*Bardwell, 2005*). However, during the course of our research, another study independently linked Crz1 to the mating response (*Carbó et al., 2017*), confirming our finding. Thus, unbiased, high-throughput exploratory analyses of protein localization changes can reveal connections that are not expected on the basis of prior knowledge.

Similarly, we identified pulsatile dynamics in the transcription factor Rtg3 in standard media in this study, and confirmed that these dynamics were altered by rapamycin. Our results contradict a previous proteome-wide screen that suggested that Rtg3 does not pulse (*Dalal et al., 2014*). However, the highly variable dynamics of pulsing from cell-to-cell (*Jacquet et al., 2003*; *Dalal et al., 2014*) and the large number of proteins independently tested in the screen, as well as differences in microscopy and imaging conditions, are plausible reasons for the discrepancy. In contrast to the time-lapse movie data used to analyse pulsing, our image data is much less comprehensive, consisting of still images of just three coarse time-points. Rather, our discovery was powered by associating protein behavior, by observing that Rtg3 had a protein change profile similar to those of other pulsatile transcription factors whose dynamics were affected by rapamycin in similar ways. That we could make findings missed by stronger experimental approaches demonstrates that looking for protein properties by association can be powerful.

As a method that exploits high-throughput imaging data, our results must be interpreted with the understanding that false positives and negatives may emerge from various steps in our data acquisition and analysis. First, occasional contamination and failures in the transformation process may occur. In this work, we found some proteins with bimodal expression in their population of cells. Our change detection algorithm correctly detects the resulting difference in protein expression in the images, but there is a possibility that these images are of mixed genetic strains. Determining whether variability in protein expression is biological or technical from the images alone is non-trivial: a previous quality-control experiment on six proteins with cell-to-cell variability in the GFP collection indicated that only two had mixed genotypes, while two had variability despite identical genotypes and another two were inconclusive (*Handfield et al., 2015*). In addition, as the GFP collection does not yet fully cover the proteome (*Chong et al., 2015*), and as cell samples must proliferate to generate an adequate sample size, our experimental method will miss some proteins because of missing data.

Second, segmentation and other image transformations can introduce artifacts. We found a cluster of bud tip proteins whose localization was predicted to change in the α-factor perturbation (Cluster Q). On manual evaluation, we observed that our segmentation algorithm incorrectly segmented the bud tip of cells in these images as a bud cell (*Figure 2—figure supplement 3*); while these proteins do exhibit a localization change, this change was incorrectly assigned to the bud cells rather than the mother.

Third, we observed that our clustering method can introduce false positives into our clusters. We observed that some of our clusters of protein change profiles had a few protein change profiles with weaker signals (e.g. Nab2 in Cluster E in *Figure 1—figure supplement 1*). Our data analysis uses distance-based clustering methods, with decrease in performance as data dimensionality increases (*Steinbach et al., 2004*). Adaptations that deal with higher-dimensionality data may be required as the number of image screens increases.

To handle false positives, we recommend that results are validated by human evaluation of the images, because an expert evaluator can determine whether the signals found by our unsupervised

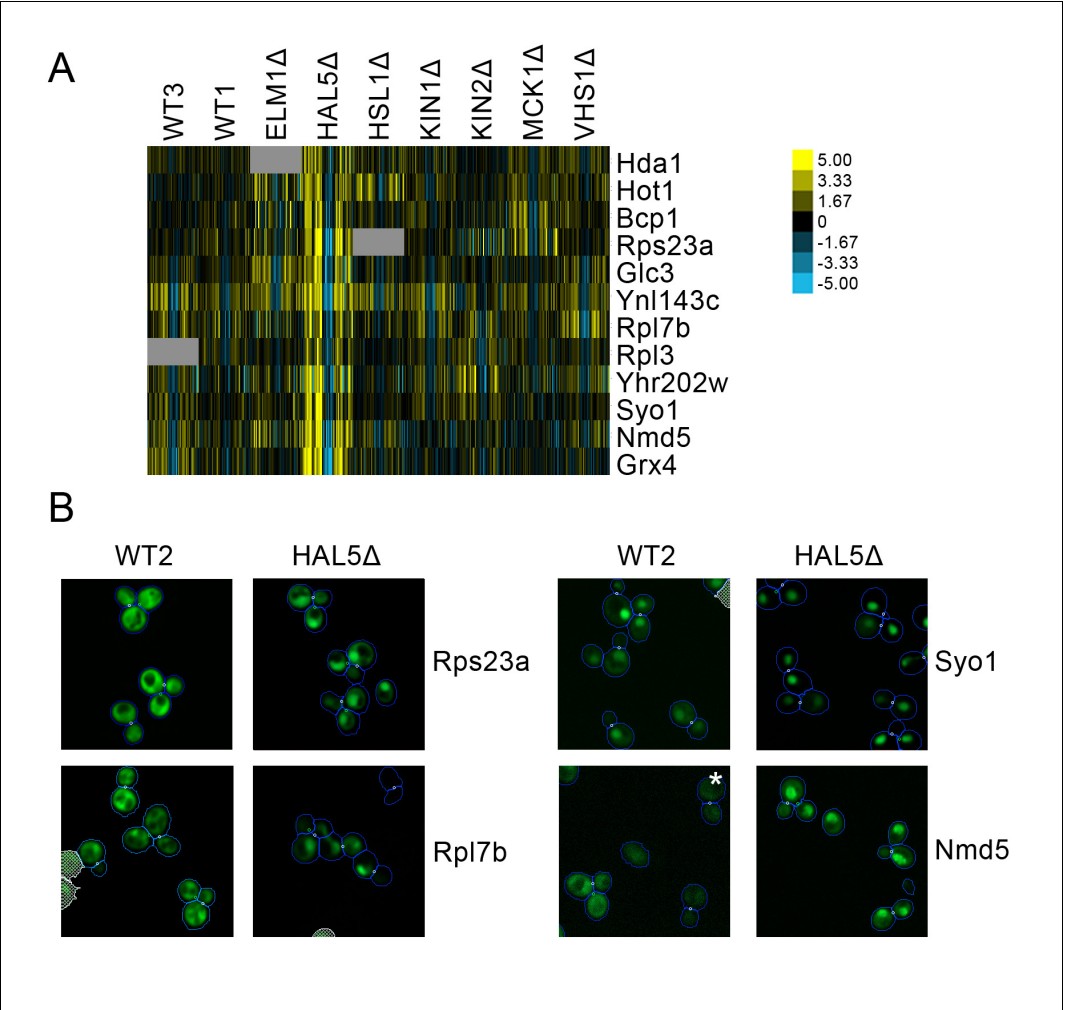

**Figure 10.** (A) Heat map showing protein change profiles, and (B) cropped micrographs for proteins in a cluster that responded specifically in *hal51Δ*. Refer to *Figure 2* for information on the heat map representation of the protein change profiles and micrograph images. In (B), we show cropped micrographs in which the labelled protein has been fused with GFP (green), and the *hal5Δ* mutant for proteins Rps23a, Rpl7b, Syo1, and Nmd5. Cropped micrographs labelled with an asterisk (*) have had their image intensities clipped at their lower range to remove noise.

DOI: https://doi.org/10.7554/eLife.31872.020

method are biological or technical. Our method complements human evaluation, which is too laborious to be applied to hundreds of thousands of images, many of which may only have subtle trends; our method drastically reduces this search space, reducing inspection to potentially only hundreds of images rather than hundreds of thousands. We focus on proteins with strong and interesting signals, equipped with knowledge of which specific screens to look at and the general nature of the putative change.

While our approach is based upon direct imaging of cells, predictive multi-perturbation approaches to study protein localization are also available (*Lee et al., 2006*, *Lee et al., 2014*; *Gardy et al., 2003*; *Horton et al., 2007*). While the latter approach is a valuable alternative when experimental data is not readily generated, we believe that predictions that are based on protein–protein interactions and mRNA expression patterns (*Lee et al., 2014*) will miss many of the localization dynamics. We found that not all clusters of protein change profiles overlap well with either of these types of datasets, suggesting that the underlying mechanisms that drive localization changes are diverse. Indeed, in recent predictions of localization change under stress (*Lee et al., 2014*),

mitochondrion to nucleus and ER to Golgi were predicted to be the most frequent type of change. We observe that transitions between the nucleus and cytoplasm were common in the localization changes that we looked at. This could be a technical effect because our change detection method is more sensitive to localization changes that reflect larger distances in our feature space (discussed in [Lu and Moses, 2016]), but we also observe that many of the transcription factors that we observed as changing under stressful conditions were not predicted to have localization changes by a previous predictive method that relied upon protein-interaction and mRNA expression datasets (Lee et al., 2014).

Integrative proteomic analyses of image screens should become increasingly possible in the future thanks to the efforts of public resources such as the Image Data Resource (Williams et al., 2016). Strategies like ours should provide the comparative context to understand the diversity of experiments contained within these databases. In medical domains, our general strategy of viewing localization changes in a multi-perturbation context may be applicable to screens of human cells. Methods are emerging for scalable GFP tagging of human proteins (Leonetti et al., 2016); given that our strategy can show protein pathway responses and compensatory rearrangements of the proteome in response to drugs, it could identify potential protein targets for knockout or combination therapies. Importantly, as our approach can separate more general responses from those more specific to a drug, it may permit selection of protein targets that minimize side effects. Furthermore, while we frame our work as relevant to perturbations, our approach can easily be extended to study proteomic differences in tissue or cell types (Uhlén et al., 2015).

## Materials and methods

### Experimental strains and image acquisition

Image data for untreated wild-type GFP-tagged yeast cells, in addition to the *rpd3Δ*, rapamycin, hydroxyurea, and α-factor perturbations, were taken from the CYCLoPs database (Koh et al., 2015).

The *iki3Δ*, *elm1Δ*, *hal5Δ*, *hsl1Δ*, *kin1Δ*, *kin2Δ*, *mck1Δ* and *vhs1Δ* strains were constructed and imaged as described by Chong et al. (2015). Fluorescent micrographs were acquired using a high-throughput spinning-disc confocal microscope (Opera, PerkinElmer) with a water-immersion 60X objective (NA 1.2, image depth 0.6 μm and lateral resolution 0.28 μm). Acquisition settings included using a 405/488/561/640 nm primary dichroic, a 568 nm detector dichroic, a 520/35 nm filter in front of camera 1 (12-bit CCD) and a 600/40 nm filter in front of camera 2 (12-bit CCD). Excitation was conducted using 488 nm (blue) and 561 nm (green) lasers at maximum power. Eight images were acquired for each well, four in the red channel and four in the green channel with simultaneous acquisition of red and green channels (binning = 1, focus height 2 μm) and an 800 ms exposure time per site.

For images that we present in this study, we cropped a 300 × 300 pixel box from the respective images showing representative cells for the sample. These images are outputted by our single cell segmentation program (Handfield et al., 2013), and show blue lines around the cells indicating the results of the segmentation, and white circles connecting mother and bud cells. Cells that were considered to be artifacts are indicated by white crossed-out regions in the images. These images were rescaled in contrast using the highest and lowest intensity pixels in the crop to better show subcellular localization patterns for proteins expressed at lower abundances.

The GFP-tagged Rtg3 strain used to evaluate nuclear localization in *Figure 4* was generated as a fusion product via homologous recombination of the native genomic sequence. Direct transformation of a linear PCR product containing codon-optimized GFP coding sequence and a selectable resistance marker flanked by gene-specific sequence yielded carboxy-terminally tagged fusion products, which were then isolated by the inferred drug resistance. To prepare strains for microscopy, strains were inoculated into synthetic minimal media and grown overnight at 30°C. Prior to imaging, stationary cultures were diluted 1/10 in fresh media and grown at 30°C for 4 hr to ensure log-phase growth and proper expression of GFP fusion products.

To produce time-lapse movies of the Rtg3 strains, cells were dosed with 200 ng/ml of rapamycin for the rapamycin treatment perturbation, and imaged at 22 °C for 16 hr at 2.5 min intervals for four z-stacks at one micrometer intervals, using a 700 ms exposure time. Cells were imaged using a Nikon spinning disk confocal microscope using a 60X oil-immersion objective. GFP excitation was at 488

nm. To analyse frames of the codon-optimized GFP strain, segmentation and tracking were conducted using Matlab on the brightfield image to identify cell peripheries. To quantify nuclear localization, we modeled the pixel intensities in each frame of the movie using a two-component generative model that uses a Gaussian distribution to model the cytoplasmic GFP signal, and a uniform distribution to model the nuclear GFP signal. Using this model, we simultaneously inferred both whether the nucleus is visible in an image, and if so, which pixels are within the nucleus. The localization score that we computed is the expected nuclear signal in each frame given these assumptions. A higher localization score indicates that the protein is distributed more to the nucleus, whereas a lower localization score indicates that the protein is distributed more to the cytoplasm. To quantify the duration of nuclear localization, we applied the 'findpeaks' function in Matlab to our localization scores.

To ensure that the consistency of protein abundance across screens was controlled, we measured the correlation of GFP-intensity across all screens. All screens have $R^2$ >0.79 with the untreated reference wild-type.

## Image analysis and quantification of localization changes

To quantify the patterns of GFP in the cells in our images, we used the single-cell segmentation and feature extraction program as described by *Handfield et al. (2013)*, specialized for the segmentation and measurement of GFP-tagged yeast cells. This method estimates the cell-cycle stage of cells using the size of the bud as a heuristic, and describes a series of 10 bins of equidistant cell-stage keypoints; for this work, we reduced the number of bins from 10 to 5 by merging adjacent bins to have more cells in each bin.

We modeled GFP-expression using quantitative, biophysically motivated features, rather than standard computer vision texture measurements (*Haralick et al., 1973*). An important property of these features is that they track the concentration and distribution of GFP-tagged proteins relative to certain cellular landmarks. As opposed to features that only track the shape of the GFP pattern, these features allow us to track relative distributions between localizations when a protein is localized to two or more compartments, and shifts the ratio of protein between these compartments.

To conduct localization change detection on these features, we applied the method that we described previously in *Lu and Moses (2016)*. This unsupervised change detection method constructs a conditional expectation for each protein and reports the direction and magnitude of deviation of the protein's change compared to this expectation; because these conditional expectations are constructed locally for each protein, differences in feature measurements between samples explained by simple variation in the morphology of an organelle or systematic biases between image screens are de-emphasized, allowing for the fairer comparison of localizations that are impacted differently by these effects. The unsupervised change detection method quantifies the putative localization change for each protein with a shared set of features. This representation encodes the nature of the localization change in the pattern across our vector of features; for instance, cytoplasm to nucleus movements are characterized by strongly positive z-scores for 'distance to cell center' and 'distance between proteins' features, which indicate that the untreated wild-type values for these features are higher than the perturbation values. Using this method, localization changes are presented in a quantitative and comparable way that summarizes each localization change as a relative measure between the untreated wild-type and the perturbed cells.

The unsupervised change detection method requires a parameterization for the number of proteins used as examples to construct a local estimate of the conditional expectation of each protein; we set this to 50 proteins, as this was shown to be optimal in our previous work. We increased the leniency of our filters for sample size reliability compared to the method described in (*Lu and Moses, 2016*), permitting vectors that have at least one cell in each bin rather than requiring at least five as originally described; this permits for the retention of more data. In practice, the lowest number of cells we observed in the reference untreated wild-type with this filter is 18, and only 13 of 3955 proteins have fewer than 25 cells. We used the more robust truncated mean profiling method described in (*Lu and Moses, 2016*). We applied the image segmentation and feature extraction method to each image screen in our dataset, and then the change detection method between each image screen paired against the WT2 screen, which served as our common reference untreated wild-type screen for all perturbations and all other untreated wild-types. A full protocol can be found at Bio-protocol (*Lu et al., 2018*).

Our unsupervised localization change detection method reports proteins that change from not being expressed (or being expressed at a level too low to confidently assign a localization) in one screen to being expressed in a recognizable localization in the other screen as localization changes. We elected to retain these signals. This strategy is consistent with previous work that has generally considered these cases to be localization changes (*Chong et al., 2015*; *Tkach et al., 2012*).

## Clustering and visualization of protein change profile

The protein change profile for individual perturbations is concatenated into a concatenated profile. We clustered these profiles using the open-source Cluster 3.0 package (*de Hoon et al., 2004*), with hierarchical agglomerative clustering with uncentered correlation distance and average linkage. This operation can be performed on the entire set of profiles, but for the heat map that we present in *Figure 2*, we reduced the number of profiles visualized to just the strongest signals prior to clustering. We required that the concatenated vector have at least three z-scores over an absolute value of 5.0 and greater than 80% of data present, resulting in 1159 proteins displayed in *Figure 2*. 46% of the proteins among the 1159 strongest localization change patterns are among the brightest 50% of proteins in the collection, indicating that the proteins with strong patterns of localization change are not simply the strongest or weakest GFP signals.

To visualize these resultant clustered matrices as heat maps, we used Java Treeview (*Saldanha, 2004*). Blue values indicate positive z-scores and yellow values indicate negative z-scores, with the intensity of the color indicating the magnitude. Grey values indicate missing data.

## Comparison with other high-throughput data sources

Enrichment analyses of the clusters in *Table 1* were conducted using the 1159 proteins visualized in *Figure 2* as the background population. We looked for gene ontology-enriched terms (*Gene Ontology Consortium, 2015*) in biological process, cellular component, and molecular function, with Benjamini-Hochberg correction (*Benjamini and Hochberg, 1995*) at a 0.05 FDR, using the YeastMine tool (*Balakrishnan et al., 2012*) on the Saccharomyces Genome Database. We report a sample of the significant terms that we found for some clusters in *Table 1*; full lists of enrichment for each cluster can be found in *Table 1—source data 1*.

To compare protein localization change profile similarity with transcript change profile similarity for these perturbations, we constructed transcript change profiles using four separate genomic expression microarray experiments for the *rpd3Δ* (*Bernstein et al., 2000*), hydroxyurea (*Dubacq et al., 2006*), rapamycin (*Hardwick et al., 1999*), and α-factor (*Roberts et al., 2000*) perturbations in yeast. We only included time points from these microarray experiments earlier or equivalent to the corresponding time points in our image screen data: this resulted in transcript data consisting of two replicates of *rpd3Δ*; three timepoints of rapamycin treatment at 15 min, 45 min, and 60 min; two timepoints of hydroxyurea treatment at 60 min and 120 min; and six timepoints of α-factor treatment at 15 min, 30 min, 45 min, 60 min, 90 min, and 120 min. Because showing all protein–protein pairs would have been prohibitive in terms of visualization (generating ~16 million points), we show protein–protein pairs formed between the proteins with the strongest changes in either protein localization changes or transcript changes. We required that a protein have either two log-fold changes above an absolute value of 2.0 in the transcript change profile, or eight values above an absolute value of 5.0 in the localization change profile. This filter resulted in 310 proteins with a transcript change, and 440 proteins with a localization change, for a total of 655 non-redundant proteins between the two lists. We then computed the correlation distance between the transcript change profile and the protein localization change profile for each pair of proteins in this list, which we visualized as the scatterplot in *Figure 7A*.

To compare protein localization change profile similarity with protein abundance change profile similarity, we constructed protein abundance change profiles using the same image data that we used to construct the protein localization change profiles. For each protein in each image screen, we measured the average intensity of the GFP signal in single cells. We calculated the difference between this measurement and that in the WT2 screen, and concatenated these differences into a profile. Similar to the transcript change profile similarity, we required that a protein have either two changes above an absolute value of 50 in the protein abundance change profile, or eight values above an absolute value of 5.0 in the localization change profile. This filter resulted in 318 proteins

with a protein abundance change, and 440 proteins with a localization change, for a total of 686 non-redundant proteins between the two datasets. We then computed the correlation distance between the protein abundance change profile and the protein localization change profile for each pair of proteins in this list, which we visualized as the scatterplot in *Figure 7B*.

Protein–protein interactions for the *Sacchromyces cerevisiae* proteome were taken from the Biogrid (*Oughtred et al., 2016*) and STRING (*Szklarczyk et al., 2017*) databases. For the Biogrid database, we compared our data with physical interactions from low-throughput experimental sources or supported by at least two PMIDs in high-throughput experimental sources. For the STRING database, we compared our data with all protein–protein associations labeled as 'binding'. We used the clusters of proteins previously defined in *Figure 2*. We counted the number of interactions in each dataset, counting interactions between the proteins of each cluster individually. To generate null expectations, we randomly shuffled all proteins within our defined clusters while retaining the sizes of each cluster, and counted the resulting number of interactions within each cluster. We repeated this simulation 10,000 times to produce a mean and standard deviation for number of interactions for randomized clusters, and report the true number of interactions for each cluster as a z-score against these expectations.

We use the manual annotations for the untreated wild-type yeast-GFP collection by *Huh et al. (2003)* to assess the distribution of untreated wild-type protein localizations within our clusters (presented in *Figure 8B*).

## Acknowledgements

We thank Helena Friesen, Taraneh Zarin, and Muluye Liku for valuable comments on the manuscript. Helena Friesen and Matej Usaj provided assistance in retrieving image datasets.

## Additional information

### Funding

| Funder | Grant reference number | Author |
| --- | --- | --- |
| National Science and Engineering Research Council | Pre-Doctoral Award | Alex X Lu |
| Canada Research Chairs | Tier II Chair | Alan M Moses |
| Canada Foundation for Innovation | | Brenda J Andrews<br>Alan M Moses |
| Canadian Institutes of Health Research | FDN-143265 | Brenda J Andrews |
| Canadian Institute for Advanced Research | Senior Fellow | Brenda J Andrews |
| Canadian Institutes of Health Research | MOP-97939 | Brenda J Andrews |

The funders had no role in study design, data collection and interpretation, or the decision to submit the work for publication.

### Author contributions

Alex X Lu, Software, Validation, Investigation, Visualization, Methodology, Writing—original draft, Writing—review and editing; Yolanda T Chong, Ian Shen Hsu, Investigation; Bob Strome, Resources, Investigation; Louis-Francois Handfield, Software, Investigation, Methodology; Oren Kraus, Conceptualization; Brenda J Andrews, Funding acquisition, Project administration, Writing—review and editing; Alan M Moses, Conceptualization, Resources, Supervision, Funding acquisition, Project administration, Writing—review and editing

### Author ORCIDs

Alan M Moses http://orcid.org/0000-0003-3118-3121

Decision letter and Author response
Decision letter https://doi.org/10.7554/eLife.31872.026
Author response https://doi.org/10.7554/eLife.31872.027

## Additional files

### Supplementary files

• Supplementary file 1. A spreadsheet containing the manual evaluations of the five clusters shown in *Figure 2*. For all proteins within each cluster, we gave to a human evaluator the images for the untreated wild-type and a perturbation predicted to change by the protein localization change profiles of the cluster. The evaluator recorded the localization in the untreated wild-type, in the perturbation, and whether a localization change was visible. If the localization change was ambiguous, the evaluator recorded why they were not able to confirm the localization change.
DOI: https://doi.org/10.7554/eLife.31872.021

• Supplementary file 2. A spreadsheet containing lists of proteins predicted to change localization under each perturbation, based upon the clusters presented in *Figure 2*.
DOI: https://doi.org/10.7554/eLife.31872.022

• Supplementary file 3. Protein localization change profiles for all kinase deletions. Columns in this spreadsheet are features, while rows are proteins.
DOI: https://doi.org/10.7554/eLife.31872.023

• Transparent reporting form
DOI: https://doi.org/10.7554/eLife.31872.024

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
