## [Decision Letter]

Thank you for submitting your article "Patterns of protein subcellular localization change inferred by integrating 100,000s of images from microscopy screens" for consideration by *eLife*. Your article has been reviewed by three peer reviewers, and the evaluation has been overseen by Emmanuel Levy as a Reviewing Editor and Naama Barkai as the Senior Editor. The reviewers have opted to remain anonymous.

The reviewers have discussed the reviews with one another, and the Reviewing Editor has drafted this decision to help you prepare a revised submission.

Summary:

Lu et al. introduce the concept of "localization profile" representing a protein's localization signature across multiple environmental conditions. The use of protein profiles has proved powerful as an approach to study protein function as well as to characterize functional relationships between proteins. It has been employed in numerous contexts and includes, for example, phylogenetic, mRNA expression, physical interactions or genetic interaction profiles.

Here, the authors characterize protein localization profiles using images from several proteome–wide fluorescence microscopy screens of the yeast GFP library. Inferring protein localization from microscopy images normally involves human intervention to train classifiers that are hardly generalizable. As a result, it has been difficult to carry out systematic and quantitative comparisons of protein localization across multiple screens carried out at different times or involving different physiological conditions. To overcome this difficulty, the authors employed an unsupervised approach to analyzing protein localization, which they applied to several proteome–wide microscopy screens.

The localization features measured across conditions formed protein localization signatures, which the authors clustered to identify groups of similarly behaved proteins. Clusters showed enrichment for functionally related proteins. Numerous changes detected were validated by known literature, and novel changes were observed. Most of the clusters of proteins with similar signature in localization–change were not associated to similar transcriptional regulation nor were they enriched for physically interacting proteins.

Essential revisions:

Globally, all reviewers found that the method shows an important potential. Numerous comments point to analyses needing to be more thorough, sometimes requiring a more systematic examination. Considering that the data underlying the kinase deletions appear to be of lower quality, I suggest cutting that part. Figures are often hard to follow, so particular attention needs to be given to the presentation of the results. Please find below a consolidated list of the reviewer's comments.

1) The authors indicate they derive descriptors even when a single cell is available. Was this decision taken based on a general optimization scheme or is it arbitrary? What fraction of the data corresponds to 1 to 5 cells? One should show that this decision brings more signal than noise.

2) A significant fraction of strains in the GFP library show fluorescence levels that are close to auto–fluorescence, and in that respect can be hard to analyze. How do such strains affect this work? What fraction of the strains in the matrix of Figure 1 are in the "top–50% abundance class?"

3) More generally, to what extent does profile similarity depend on signal intensity or protein abundance?

4) How was the number of features decided upon? Each feature described in Figure S1 has 10 pixels associated with it (and therefore I assume 10 parameters). Why are 10 features necessary when, e.g., considering "distance to the cell edge." Could one feature, or perhaps an average of the features facilitate interpretation while being equally informative in the profile comparison? It would be necessary to establish an objective criterion that can be optimized – such as "number of changes detected across different conditions versus across replicates of the same condition." (this objective criterion can also be used to set the minimum number of cells to be used, see point #1.). A detailed depiction of the features employed is needed.

5) The approach is validated using a set of 20 pictures, which may not account for global effects. A more general benchmark would, therefore, make sense in that respect too (see comment #4).

6) In the validation, the authors infer a false positive rate of ~50%. They should provide a sense of what the origin of these false positives is, and how it may impact the conclusions. For example, that proteins in the same clusters to not exhibit similar transcription profile, physical interactions, or localization.

7) An analysis of a limited number of clusters is presented in Table 1, where GO enrichment is carried out. Are these the only clusters showing functional enrichment? The global nature of this work begs for a more general analysis of GO enrichment, i.e., by defining clusters across the entire matrix and analyzing GO enrichment for all of them, not a handful only. Even if a few are described in Table 1, the results can be provided as Supplementary data. In addition, "numbers" can be added to Table 1 (number of proteins in the cluster and number corresponding to the annotation being enriched).

8) Figure 4C contains cells that do not appear to contain any green fluorescence, while others show a strong signal. Such bimodal behavior is worrying as it may not be biological. Perhaps that some cells have lost their fluorescence during the SGA process for example. Is the localization profile affected by such cases? Is such bimodal behavior widespread? If so it should be accounted for in the analyses, and such cases would need to be filtered or flagged.

9) Global "quality control" figures should be shown, suggestions for this are given below:

– Subsection “An unsupervised analysis of protein localization changes in over 280,000 images”: More details should be presented regarding this sentence. Can a distribution of "localization change" score be shown for replicates of the same condition and different conditions? More of such "sanity check" plots would help get a better sense of how the method is doing. (See comment #1)

– The consistency of protein abundance (or rank abundance) across the different screen should be controlled.

– The association between change in localization, expression–change, and PPIs is shown in Figure 6 for a few cases, but it would be more valuable to show it in a systematic fashion, for example by showing (i) how Z–score for profile change in the data from Figure 1 correlate with a Z–score for abundance change, or (ii) How profile similarity relates to other measures of functional relatedness? E.g., protein interaction profile similarity, genetic interaction profile similarity, expression–profile similarity, etc. Such a bird's eye view of the data is lacking and would add much to this work.

– The last analysis suggests that proteins with different localizations can exhibit a similar change. This is counter–intuitive, and more data could be provided to examine the origin of such cases

– Looking at Figure 6C, why is cluster I, which contains ribosomal proteins, not enriched in physical interactions? If using BioGrid one does not need to limit oneself to low–throughput data. A large and high–quality dataset can be obtained by taking all physical interactions supported by more than one PMID. Lastly, the "clusters circles" of Figure 6C do not overlay with points.

10) The authors identified Rtg3 to be a pulsing transcription factor through cluster analysis and conducted time–lapse movies to confirm such dynamics. In the cited reference (Dalal et al., 2014), pulsing was defined as TFs that exhibit dynamic shuttling between nucleus and cytoplasm during the steady–state response to added stresses. In Figure 3, the authors didn't indicate when the stress was added and whether the cells on average have reached steady–state. The authors didn't specify what stress condition each cell represent in Figure 3. Lastly, to analyze "longest pulse duration" (Figure 3C) does not provide a robust measure. The mean/median duration should be presented.

11) Analysis of protein level change. In Figure 6A, the authors compared localization pattern with microarray pattern. However, it would be more natural to compare the protein expression pattern with microarray pattern, and more importantly protein expression with localization changes. It is indeed always difficult to merge different data from different laboratories, but protein expression is readily available from the images as the sum of GFP intensities in the cell.

---

## [Author Response]

Essential revisions:Globally, all reviewers found that the method shows an important potential. Numerous comments point to analyses needing to be more thorough, sometimes requiring a more systematic examination. Considering that the data underlying the kinase deletions appear to be of lower quality, I suggest cutting that part. Figures are often hard to follow, so particular attention needs to be given to the presentation of the results. Please find below a consolidated list of the reviewer's comments.1) The authors indicate they derive descriptors even when a single cell is available. Was this decision taken based on a general optimization scheme or is it arbitrary? What fraction of the data corresponds to 1 to 5 cells? One should show that this decision brings more signal than noise.

We believe that this comment was based on a lack of clarity in the description of our previously published methods. We require at least one mother–bud pair in each of the 10 bins that we define, which track mother and bud cells separately over 5 stages in the cell cycle. This means that we require a minimum of 10 cells per sample, distributed across the cell cycle. In practice, the lowest number of cells we observe in the reference untreated wild–type with this filter is 18, and only 13 of 3955 proteins have fewer than 25 cells.

We have edited subsection “An unsupervised analysis of protein localization changes in over 280,000 images” and included a new figure (Figure 1) to better explain our image analysis pipeline and included the details about the numbers of cells in subsection “Image Analysis and Quantification of Localization Changes”. We hope that this is clearer in the revised manuscript.

These decisions are based on the desire to avoid filtering out proteins from the analysis because of missing data as we add more screens to the analysis. We believe that including as many proteins as possible is important because we wish to provide as much as possible a proteome–scale analysis. In addition, for technical reasons, our localization change detection algorithm (Lu and Moses, 2016) actually works better when we have more proteins in the analysis. For example, changing the minimum requirement from 1 to 5 cells per bin filters out an additional 730 (18.4%) proteins in the reference untreated wild–type.

2) A significant fraction of strains in the GFP library show fluorescence levels that are close to auto–fluorescence, and in that respect can be hard to analyze. How do such strains affect this work? What fraction of the strains in the matrix of Figure 1 are in the "top–50% abundance class?"

Our localization change algorithm automatically handles low intensity proteins where the GFP signal is not recognizable. Low intensity proteins show a distinctive pattern in our feature space (see the cluster labelled as “ambiguous” in Figure 6 of Handfield et al., 2013) and can be handled identically to proteins with a distinct localization: If the protein remains low intensity in the perturbation, the protein will have the same pattern of features and will not be identified as having a localization change.

On the other hand, if proteins change from being low intensity in the wild–type to having a recognizable localization in the perturbation (or vice versa), these will be identified as having a localization change. We elected to retain these signals. There is some ambiguity on whether these changes are “truly” localization changes or instead should be thought of as “abundance” changes, but we believe that this shows that the distinction between “abundance” and “localization” is somewhat arbitrary.

In addition, we believe our perspective is in line with previous localization change detection work has generally considered these cases to be localization changes: Tkach et al., (2012) includes localization changes where the protein abundance is at auto–fluorescence levels in untreated cells, Chong et al., (2015) also includes localization changes of this nature. We have added a brief discussion of abundance and localization changes to the revised manuscript to subsection “Image Analysis and Quantification of Localization Changes”.

For all proteins that we analyzed in this study, 50% of proteins have a protein abundance in the reference untreated wild–type greater than 41.0 A.U. For the 1159 proteins that we show in Figure 2, 529 (or 45.6%) of proteins have a protein abundance greater than 41.0 A.U. We have added these numbers to subsection “Clustering and Visualization of Protein Change Profile.”

3) More generally, to what extent does profile similarity depend on signal intensity or protein abundance?

For most screens, the strongest localization change signals do not co–occur with a significant abundance change, and the strongest abundance changes do not co–occur with a significant localization change. We have added Figure 7—figure supplement 1, which shows scatterplots of the protein localization change profile magnitude compared to the average protein abundance change for some screens.

The *rpd3*Δ perturbation is an exception. This screen shows more localization changes that co–occur with a strong change in abundance. These results are consistent with previous work; while Chong et al., (2015) report little correlation between abundance and localization changes for the hydroxyurea and rapamycin perturbations, 17 of the 31 localization changes that they detect for the *rpd3*Δ perturbation also significantly change in abundance.

We have reported these findings in subsection “Cluster associations found by our unsupervised exploration of localization changes are complementary to other high–throughput experiments” of the revised manuscript.

4) How was the number of features decided upon? Each feature described in Figure S1 has 10 pixels associated with it (and therefore I assume 10 parameters). Why are 10 features necessary when, e.g., considering "distance to the cell edge." Could one feature, or perhaps an average of the features facilitate interpretation while being equally informative in the profile comparison? It would be necessary to establish an objective criterion that can be optimized – such as "number of changes detected across different conditions versus across replicates of the same condition." (this objective criterion can also be used to set the minimum number of cells to be used, see point #1.). A detailed depiction of the features employed is needed.

Once again, we believe there is confusion due to the lack of clarity in the description of the methods in the submitted version of the manuscript.

The 10 features result from the separation our cells into 10 bins (5 cell cycle stages for the bud and 5 cell cycle stages for the mother cells). The features shown for proteins are an average of the single cells in each bin. The repeated features in the protein localization change profile are designed to show if the protein localization change is dependent on cell cycle and/or on mother/bud cell type. We have clarified this in the revised manuscript (Figure 1).

Feature selection and optimization are key areas for research in automated microscopy image analysis. It is likely that they could lead to easier interpretation of localization changes. However, since our current manuscript is focused on data integration from multiple screens, we believe that it is outside the scope to begin consideration of these challenging data analysis problems.

5) The approach is validated using a set of 20 pictures, which may not account for global effects. A more general benchmark would, therefore, make sense in that respect too (see comment #4).

We take the point that 20 pictures were not convincing. We have therefore expanded our manual validation set to 80 pairs of images, recorded in Supplementary Data 2. We analyzed the images in Clusters E, F, J, Q, and S, shown in Figure 1. For each set of images, we have reported whether a localization change was visible to the human observer, and quantified false positive rates with respect to these observations.

We thank the reviewer for the suggestion to develop a general benchmark for protein localization changes under perturbations. This would be useful to the community to develop new statistical approaches. The images that we have manually inspected as a part of this study could form the seed for such a data resource, and we have made them available in Supplementary Data 2. However, we believe that further development of this resource is beyond the scope of the current manuscript.

6) In the validation, the authors infer a false positive rate of ~50%. They should provide a sense of what the origin of these false positives is, and how it may impact the conclusions. For example, that proteins in the same clusters to not exhibit similar transcription profile, physical interactions, or localization.

To address the origin of the false positives, we used the set of manually inspected images (discussed in response to point #5). We found at least three causes for false positives: (1) protein change profiles with weaker signals that are clustered among the profiles with stronger signals. These are true “statistical” false positives where the noise passes the threshold that we chose and happens to fall in a pattern similar enough to the strong signals. (2) localization changes are too subtle, complex, or borderline to identify consistently by human eye. These often include cases where only a subset of cells in the image show the localization change, and/or the localization changes in in the quantitative appearance of an organelle. While these are false positives in the sense that a human observer would not call them localization changes, it is likely that they are simply subtle biological effects. (3) technical issues, particularly related to mis–segmentation of cells with previously unseen morphology. We believe that our methods to a surprisingly good job at dealing with changes in cell morphology, but they are not perfect.

A table of all our manual annotations can be found in Supplementary Data 2, and images of these proteins are available on CYCLoPS. We have added some supplementary figures to address this question and discussed all of these cases along with specific examples and images in the revised manuscript. Figure 2—figure supplement 1 shows examples of case 1 and Figure 2—figure supplement 2 shows examples of case 2; we discuss these in the result section entitled “An unsupervised analysis of protein localization changes in over 280,000 images”. Supplementary Data 5 shows an example of case 3, and we discuss this as a source of false positives in the Discussion section.

To address whether the results of the global comparisons with other data could be explained by the presence of false positives, we examined cases where localization change patterns did not co–occur with transcript or protein abundance patterns, or with protein interactions and shared subcellular localization, in the context of the clusters that we had manually identified the false positives. We found that the false positives could not completely explain the lack of co–occurrence for any cluster. We report these results in the result section entitled “Cluster associations found by our unsupervised exploration of localization changes are complementary to other high–throughput experiments” with reference to Figure 2—figure supplement 2; we believe that verification in this limited set is the best we can do to eliminate the possibility that our global results are explained by false positives.

7) An analysis of a limited number of clusters is presented in Table 1, where GO enrichment is carried out. Are these the only clusters showing functional enrichment? The global nature of this work begs for a more general analysis of GO enrichment, i.e., by defining clusters across the entire matrix and analyzing GO enrichment for all of them, not a handful only. Even if a few are described in Table 1, the results can be provided as sup. Data. In addition, "numbers" can be added to Table 1 (number of proteins in the cluster and number corresponding to the annotation being enriched).

We have added the numbers of proteins in the table as requested. We have now included any significant GO terms for all of the manually identified clusters in Source Data 1.

Presenting GO enrichments for automatically chosen clusters could be misleading because global effects (such as changes in morphology, growth or imaging conditions, etc.) that differentially effect proteins with different subcellular localization patterns might still be significant. For example, clusters showing differences between untreated wild–type screens show strong GO enrichments. We have therefore decided to stick with manually identified clusters, staying true to the use of clustering as exploratory data analysis to identify patterns.

8) Figure 4C contains cells that do not appear to contain any green fluorescence, while others show a strong signal. Such bimodal behavior is worrying as it may not be biological. Perhaps that some cells have lost their fluorescence during the SGA process for example. Is the localization profile affected by such cases? Is such bimodal behavior widespread? If so it should be accounted for in the analyses, and such cases would need to be filtered or flagged.

We agree that some of our detected localization changes may be due to contamination or technical issues in the transformation: occasional errors are inevitable in high–throughput screens. We have discussed contamination as a source of false positives in the Discussion section of the revised manuscript. We note that this issue is a limitation of the GFP collection. In these cases, our analysis correctly identifies that there is a localization change in the images.

Unfortunately, it is not possible to confirm from the images alone if bimodal behavior is caused by a technical issue. In a previous study, we performed quality control experiments on 6 proteins with highly variable expression in the GFP collection indicated that only 2 had mixed genotypes, while 2 had variability despite identical genotypes, and another 2 were inconclusive (Handfield et al., 2014).

We therefore decided not to filter bimodally–expressed proteins because this may unnecessarily exclude real and interesting biological findings. Instead, we advocate generation of independent GFP–tagged strains to verify the localization change, as we have done for our Rtg3 experiment. Indeed, the specific case of some ribosomal subunits being expressed at very low levels in some cells in the *iki3*Δ perturbation needs further confirmation and we have removed these examples from our results in subsection “Stress–responsive transcription factors that exhibit stochastic pulsatility may be controlled by the same mechanism”.

9) Global "quality control" figures should be shown, suggestions for this are given below:– Subsection “An unsupervised analysis of protein localization changes in over 280,000 images”: More details should be presented regarding this sentence. Can a distribution of "localization change" score be shown for replicates of the same condition and different conditions? More of such "sanity check" plots would help get a better sense of how the method is doing. (See comment #1)

We thank the reviewers for this useful comment. We have now included a comparison of localization change profiles for replicates as Supplementary Data 1 and discuss these results in subsection “An unsupervised analysis of protein localization changes in over 280,000 images”. We analyse the correlation between the *rpd3*Δ and the *iki3*Δ replicates, as well as comparisons to different conditions. We find that the replicates exhibit strong correlation, particularly for proteins that show strong signals, while different perturbations are poorly correlated.

– The consistency of protein abundance (or rank abundance) across the different screen should be controlled.

We compared the protein abundance across all the screens and we find R2 > 0.79 with the untreated wildtype. We have reported this result in subsection “Experimental Strains and Image Acquisition”.

– The association between change in localization, expression–change, and PPIs is shown in Figure 6 for a few cases, but it would be more valuable to show it in a systematic fashion, for example by showing (i) how Z–score for profile change in the data from Figure 1 correlate with a Z–score for abundance change, or (ii) How profile similarity relates to other measures of functional relatedness? E.g., protein interaction profile similarity, genetic interaction profile similarity, expression–profile similarity, etc. Such a bird's eye view of the data is lacking and would add much to this work.

We thank the reviewers for this excellent suggestion. The suggested analysis of comparing profile similarity better demonstrates the point that we were trying to make in this section, which was that the associations that we found through clustering protein localization change profiles could not entirely be found by clustering other similar high–throughput datasets. We have now included the suggested analysis as Figure 7, which compares protein localization change profile similarity with both protein abundance change, and transcriptional profile similarity, in the revised manuscript.

– The last analysis suggests that proteins with different localizations can exhibit a similar change. This is counter–intuitive, and more data could be provided to examine the origin of such cases

Because our analysis captures relative movements in protein localization, it does not depend upon the protein sharing a localization in the untreated wild–type. For example, we show that a subset of proteins moved to the vacuole under rapamycin treatment in subsection “Membrane proteins exhibit a reciprocal pattern of localization changes following hydroxyurea or rapamycin treatment”, despite proteins showing different localizations in the untreated wild–type. We have now provided a clearer representation of this phenomenon using pie charts in Figure 8B, and discussed the origin of heterogeneity for three clusters, S, J and T in the text.

– Looking at Figure 6C, why is cluster I, which contains ribosomal proteins, not enriched in physical interactions? If using BioGrid one does not need to limit oneself to low–throughput data. A large and high–quality dataset can be obtained by taking all physical interactions supported by more than one PMID. Lastly, the "clusters circles" of Figure 6C do not overlay with points.

We thank the reviewer for pointing out this problem.

We have expanded the enrichment analysis for BIOGRID to incorporate all physical interactions supported by more than one PMID. We note that this does not change the number of interactions substantially: we originally analyzed 47,498 non–redundant low–throughput physical interactions and incorporating high–throughput physical interactions supported by more than one PMID only adds 7,656 more interactions.

Upon closer analysis, we found that BIOGRID contains very few interactions between ribosomal subunits: we found only 41 physical interactions between ribosomal subunits even before filtering for redundancy. We therefore decided to also check for enrichment for protein associations that had a “binding” interaction type in the STRING database. The ribosomal clusters are now enriched with the STRING database, but other clusters have lower enrichment in STRING compared to BIOGRID. In the revised manuscript, we show enrichment in both databases side–by–side in Figure 8A.

As we are unable to explain these inconsistencies in BIOGRID and STRING, we have downgraded our claims about protein–protein interactions. Instead of stating that our clusters are not enriched for physical protein–protein interactions in general, we state that we cannot find enrichment in these databases.

Finally, figure 6C no longer appears in the revised manuscript.

10) The authors identified Rtg3 to be a pulsing transcription factor through cluster analysis and conducted time–lapse movies to confirm such dynamics. In the cited reference (Dalal et al., 2014), pulsing was defined as TFs that exhibit dynamic shuttling between nucleus and cytoplasm during the steady–state response to added stresses. In Figure 3, the authors didn't indicate when the stress was added and whether the cells on average have reached steady–state. The authors didn't specify what stress condition each cell represent in Figure 3. Lastly, to analyze "longest pulse duration" (Figure 3C) does not provide a robust measure. The mean/median duration should be presented.

We have the edited the subsection “Stress–responsive transcription factors that exhibit stochastic pulsatility may be controlled by the same mechanism” to clarify this.

We have now indicated when stress (Rapamycin treatment) was added in the caption of Figure 4. Since we observe pulsing in standard media (unstressed cells) over many hours, we are confident that these cells have reached steady state, and that pulsing occurs at steady–state.

Our finding is that Rapamycin alters the pulsing behavior by inducing accumulation of protein in the nucleus, and we have clarified this in the new manuscript. Our observations suggest that the pulsatile dynamics of Rtg3 are similar to Msn2, which also exhibits pulsing under many constant conditions, but accumulates in the nucleus under rapamycin treatment.

Finally, we have now presented an analysis that compares the average duration of nuclear localization, instead of the largest pulse in Figure 4C and find similar results. In order to do so, we needed to reliably identify all the pulses in each trace, and we therefore devised a more robust method for estimating nuclear localization. We describe this briefly in the Materials and methods section; the traces in Figure 4 have been updated with measurements using this new method.

11) Analysis of protein level change. In Figure 6A, the authors compared localization pattern with microarray pattern. However, it would be more natural to compare the protein expression pattern with microarray pattern, and more importantly protein expression with localization changes. It is indeed always difficult to merge different data from different laboratories, but protein expression is readily available from the images as the sum of GFP intensities in the cell.

We have now incorporated protein expression into our analyses. We now show a comparison of protein abundance profile similarity to protein localization similarity in Figure 7 and discussed the results in the test. Our cluster–by–cluster comparison of protein localization change profiles also now shows both the transcript and the protein abundance profiles (Figure 2—figure supplement 1 and Figure 2—figure supplement 2).

We have also included scatterplots comparing the magnitude of the protein localization change profile with the protein abundance change for some screens in Figure 7—figure supplement 1. For most screens, the strongest localization change signals do not co–occur with a significant abundance change, and the strongest abundance changes do not co–occur with a significant localization change.